# TS-LIF: A Temporal Segment Spiking Neuron Network for Time Series Forecasting

**Shibo Feng**[1,4]*, **Wanjin Feng**[2]*, **Xingyu Gao**[2], **Peilin Zhao**[3]†, **Zhiqi Shen**[1]†

shibo001@ntu.edu.sg; {fengwanjin,gaoxingyu}@ime.ac.cn
Nanyang Technological University[1]   University of Chinese Academy of Sciences[2]
Tencent AI Lab[3]   Webank-NTU Joint Research Institute on Fintech, NTU, Singapore[4]

## Abstract

Spiking Neural Networks (SNNs) offer a promising, biologically inspired approach for processing spatiotemporal data, particularly for time series forecasting. However, conventional neuron models like the Leaky Integrate-and-Fire (LIF) struggle to capture long-term dependencies and effectively process multiscale temporal dynamics. To overcome these limitations, we introduce the Temporal Segment Leaky Integrate-and-Fire (TS-LIF) model, featuring a novel dual-compartment architecture. The dendritic and somatic compartments specialize in capturing distinct frequency components, providing functional heterogeneity that enhances the neuron's ability to process both low- and high-frequency information. Furthermore, the newly introduced direct somatic current injection reduces information loss during intra-neuronal transmission, while dendritic spike generation improves multi-scale information extraction. We provide a theoretical stability analysis of the TS-LIF model and explain how each compartment contributes to distinct frequency response characteristics. Experimental results show that TS-LIF outperforms traditional SNNs in time series forecasting, demonstrating better accuracy and robustness, even with missing data. TS-LIF advances the application of SNNs in time-series forecasting, providing a biologically inspired approach that captures complex temporal dynamics and offers potential for practical implementation in diverse forecasting scenarios. The source code is available at https://github.com/kkking-kk/TS-LIF.

## 1 Introduction

Spiking Neural Networks (SNNs) have garnered significant attention due to their biological plausibility and unique capacity to process spatiotemporal information (Hu et al., 2024). Unlike traditional artificial neural networks (ANNs), which rely on continuous activations, SNNs utilize discrete spikes as their primary communication mechanism (Wang et al., 2024). This event-driven nature allows SNNs to operate efficiently, only processing information when necessary, making them highly suited for tasks involving sparse, time-dependent data (Gast et al., 2024). By encoding information through the precise timing of spikes, SNNs achieve fine temporal resolution, providing a significant advantage in applications requiring both temporal and spatial accuracy (Zhu et al., 2024). Moreover, the asynchronous processing of SNNs closely mimics biological neurons, enabling energy-efficient computation (Ganguly et al., 2024; Bellec et al., 2020).

One domain that aligns naturally with SNNs is time series forecasting, which involves predicting future values based on historical observations. It is critical in various domains, including finance, weather prediction, healthcare, and energy monitoring (Lin et al., 2024; Ilbert et al., 2024a; Angelopoulos et al., 2024). The sequential nature of time series data, characterized by time dependencies, aligns well with SNNs' temporal processing abilities. Traditional deep learning models, such as Temporal Convolutional Networks (TCNs), and Transformer, have been effective in capturing long-term dependencies (Luo & Wang, 2024; Wu et al., 2021; Ilbert et al., 2024b). However,

---

*Equal contribution.
†Corresponding to: Peilin Zhao (masonzhao@tencent.com), and Zhiqi Shen (zqshen@ntu.edu.sg)

these models typically require significant computational resources to manage complex temporal relationships (Liu et al., 2023; Feng et al., 2025). In contrast, SNNs, with their event-driven and sparse computational architecture, can offer a more efficient solution, particularly for applications that involve sparse temporal events and demand low energy consumption (Lv et al., 2024).

Despite the potential benefits, applying SNNs to time series forecasting has been limited. A notable exception is the work by Lv et al. (2024), which demonstrated that SNNs could achieve competitive results in this domain, particularly in terms of efficiency when implemented on neuromorphic hardware. However, broader adoption remains constrained by the limitations of the widely used Leaky Integrate-and-Fire (LIF) neuron model. While the LIF neuron is biologically plausible and computationally efficient, its rapid membrane potential decay impairs its ability to capture long-term dependencies (Wang & Yu, 2024). Furthermore, it struggles to process multi-timescale information, which is crucial for understanding both short-term fluctuations and long-term trends (Zheng et al., 2024). These challenges restrict the effectiveness of LIF-based SNNs in complex forecasting tasks, where accurate predictions require capturing patterns across multiple temporal scales.

To address these limitations, we propose the Temporal Segment LIF (TS-LIF) Neuron Model, specifically designed for time series forecasting. TS-LIF incorporates a dual-compartment mechanism to process information across different timescales. This model extends the standard LIF neuron by introducing dendritic and somatic compartments, each responsible for capturing distinct frequency components of the input signal. Furthermore, the addition of direct somatic current injection mitigates information loss during intra-neuronal transmission, while dendritic spike generation improves the neuron's capacity for extracting information across multiple scales. We establish the stability conditions for this dual-compartment model and derive the frequency-domain transfer functions for both the dendritic and somatic compartments. The TS-LIF model was evaluated on four benchmark datasets using CNN, RNN, and Transformer architectures, consistently outperforming previous LIF-based models. Additionally, TS-LIF demonstrates a significant advantage in maintaining accuracy under scenarios of missing inputs. Finally, through ablation studies, we show how TS-LIF achieves superior performance by effectively decomposing input signals into different frequency components. In summary, our contributions include:

- We propose the Temporal Segment LIF (TS-LIF) model, a dual-compartment neuron with dendritic and somatic branches that effectively captures multi-scale temporal features, enhancing time series forecasting.

- We establish stability conditions for TS-LIF, ensuring robustness, and derive frequency-domain transfer functions that illustrate the distinct contributions of dendritic and somatic compartments to temporal processing.

- We validate TS-LIF on four benchmark datasets using CNN, RNN, and Transformer architectures, demonstrating consistent improvements over LIF-based SNNs, superior accuracy, and robustness to missing inputs.

## 2 RELATED WORK

### 2.1 MODELING LONG-TERM DEPENDENCIES IN SNNS

Early SNN research using the LIF neuron model focused on simulating neuronal dynamics but struggled with long-term dependencies, restricting its effectiveness in memory-intensive tasks.(Wang et al., 2023). To address this challenge, several advanced neuron models have been proposed.

The Gated Leaky Integrate-and-Fire (GLIF) model introduced a gating mechanism to regulate temporal information flow, improving long-term sequence modeling while maintaining energy efficiency (Yao et al., 2022). Similarly, Wang & Yu (2024) explored autaptic connections to enhance long-term dependency processing. Dual-compartment models have also shown promise in improving memory retention. The Two-Compartment LIF (TC-LIF) model divides memory between dendritic and somatic compartments, enhancing gradient propagation and improving long-sequence retention (Zhang et al., 2024). Building on this, the Learnable Multi-hierarchical (LM-H) neuron model introduced learnable parameters to dynamically balance historical and current information (Hao et al., 2024). Additionally, Zheng et al. (2024) proposed a multi-compartment neuron model with temporal dendritic heterogeneity, enabling neurons to process different time-scale inputs.

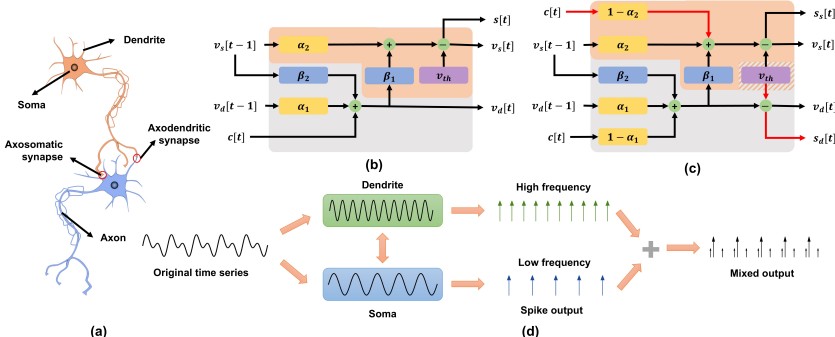

Figure 1: Diagram of Neuronal Signal Processing and Integration: (a) Structural organization of neuronal signal transmission, highlighting axosomatic and axodendritic synapses. (b) A generalized two-compartment spiking neuron model, applicable to TC-LIF or LM-H models, with dendritic (gray) and somatic (orange) compartments. (c) Proposed TS-LIF model with the newly introduced direct somatic current injection and dendritic spike generation (highlighted in red). (d) Time series decomposition and spike output generation in the TS-LIF model.

Despite these advances, current models still fall short in effectively decomposing and integrating features from different time scales within a single input signal, leaving room for further exploration in this area.

## 2.2 TIME SERIES FORECASTING

Time series refers to a sequence of data points recorded over time intervals, which is crucial for understanding temporal dynamics in various fields (Wu et al., 2020; Feng et al., 2024; Woo et al., 2024). While recent advancements in model architectures have improved forecasting accuracy, balancing performance with computational efficiency remains a challenge.

Traditional models like Recurrent Neural Networks (RNNs), including LSTMs and GRUs, are widely used for sequential data but often struggle with long-term dependencies and inefficiencies on large datasets (Ilhan et al., 2021; Dera et al., 2023). Temporal Convolutional Networks (TCNs) offer a more scalable alternative by capturing long-range dependencies through dilated convolutions, allowing for parallel processing of sequences (Lea et al., 2017; Luo & Wang, 2024). Transformers, initially designed for natural language processing, have also been adapted for time series forecasting and generation (Ilbert et al., 2024a; Zhang & Yan, 2023; Zhicheng et al., 2024). Their self-attention mechanism models long-range dependencies effectively, while variants like Informer and Autoformer use sparse attention and decomposition techniques to reduce computational demands (Zhou et al., 2021; Wu et al., 2021).

However, the computational requirements of TCNs and Transformers, particularly regarding energy consumption, remain substantial. Implementing these models in resource-constrained environments is still challenging, even with optimizations such as sparse attention and hybrid approaches.

## 3 PRELIMINARIES

### 3.1 TIME SERIES PROBLEM SETTING

We consider the task of multivariate time series forecasting, where the observations are represented as a sequence $\mathbf{X} = \{\mathbf{x}_1, \mathbf{x}_2, \ldots, \mathbf{x}_T\} \in \mathbb{R}^{T \times C}$. Here, $T$ represents the number of time steps, and $C$ denotes the number of variables. The goal is to learn a predictive function $f$ that generates future values $\mathbf{Y} = \{\mathbf{x}_{T+1}, \mathbf{x}_{T+2}, \ldots, \mathbf{x}_{T+L}\} \in \mathbb{R}^{L \times C}$ for the next $L$ time steps.

To achieve this, time series decomposition can be utilized to reveal features across different temporal scales, such as short-term variations and long-term trends. These features play a crucial role in developing models that can handle both rapid fluctuations and slower patterns in the data. By

modeling these components individually or jointly, the predictive function $f$ can better capture the underlying structure of the time series, thereby enhancing forecasting accuracy.

## 3.2 LIF Neuron Model

The Leaky Integrate-and-Fire (LIF) neuron is widely used in SNNs due to its computational simplicity and biological relevance. The evolution of the membrane potential in the discrete-time domain is expressed as:

$$v[t] = \alpha v[t-1] + c[t] - v_{\text{th}} s[t-1], \tag{1}$$

where $\alpha < 1$ is the decay factor, $v[t]$ represents the membrane potential at time step $t$, and $c[t]$ is the input current. The term $v_{\text{th}} s[t-1]$ accounts for resetting the membrane potential after a spike. The spike output is determined by the Heaviside step function $H(\cdot)$, given by:

$$s[t] = H(v[t] - v_{\text{th}}), \tag{2}$$

where $H(x)$ outputs 1 if $x \geq 0$ and 0 otherwise. This function indicates whether the membrane potential has exceeded the threshold $v_{\text{th}}$, thereby triggering a spike.

Assuming the initial membrane potential $v[0] = 0$, the evolution of the membrane potential simplifies to:

$$v[t+1] = \sum_{k=1}^{t+1} \alpha^{t-k+1} c[k] - v_{\text{th}} \sum_{k=1}^{t} \alpha^{t-k} s[k]. \tag{3}$$

This equation shows that $v[t+1]$ is a weighted sum of past input currents $c[k]$ and spike output $s[k]$, with older values decaying exponentially (Wang & Yu, 2024). The LIF neuron integrates inputs over a short window, acting like a low-pass filter. Consequently, it primarily responds to recent inputs, making it less effective in capturing long-term dependencies, which limits its use in tasks that require extended memory and complex spatiotemporal processing.

## 3.3 Dendrites and Soma

Biological neurons are often represented with a dual-compartment architecture, comprising dendrites and soma, each handling specific signal processing functions, as illustrated in Figure 1(a). Dendrites receive synaptic inputs and integrate them over time, while the soma acts as the central decision-making unit, determining whether to generate a spike based on the accumulated inputs, as shown in Figure 1(b) (Zhang et al., 2024; Hao et al., 2024). The dynamics of this system are mathematically described as:

$$\begin{cases} v_d[t] = \alpha_1 v_d[t-1] + \beta_1 v_s[t-1] + c[t], \\ v_s[t] = \alpha_2 v_s[t-1] + \beta_2 v_d[t] - v_{\text{th}} s_s[t-1], \\ s_s[t] = H(v_s[t] - v_{\text{th}}). \end{cases} \tag{4}$$

Here, $v_d[t]$ and $v_s[t]$ represent the membrane potentials at the dendrites and soma, respectively. $\alpha_1$ and $\alpha_2$ are decay factors that modulate the influence of previous membrane potentials, while $\beta_1$ and $\beta_2$ signify cross-compartmental interactions. The soma generates a spike when its membrane potential $v_s[t]$ exceeds the threshold $v_{\text{th}}$.

By making minor adjustments to Equation (4), the TC-LIF and LM-H neuron models can be derived. These models can, to some extent, mitigate the problem of vanishing gradients by appropriately tuning the decay factors $\alpha$ and cross-compartmental interactions $\beta$, enabling them to better model long-term dynamic relationships. However, these models lack a clear distinction in processing capabilities across different temporal scales, such as high and low frequencies, between the dendritic and somatic compartments. Moreover, there is no theoretical validation of their robustness or temporal processing capabilities, limiting their effectiveness in scenarios requiring explicit multi-timescale feature extraction and reliable long-term memory retention.

## 4 Methodology

### 4.1 Temporal Segment LIF Neuron

In the dual-compartment model described by Equation (4), the input current $c[t]$ flows through the dendrites before reaching the soma, which can result in information loss during transmission. For

example, if the dendritic compartment $v_d$ primarily captures the low-frequency features of $c[t]$, it becomes challenging for the somatic compartment $v_s$ to recover the original high-frequency components without direct current input. This limitation highlights the model's difficulty in effectively handling multi-scale information simultaneously.

Biological neurons contain numerous synapses that function as independent pattern detectors (including identity mapping), ensuring soma receive sufficient information (Hawkins & Ahmad, 2016). However, describing the complete neural dynamics using simple formulas is impractical. To address this, we propose the TS-LIF neuron model, which incorporates shortcut mechanisms. These mechanisms can be viewed as dendritic pathways performing identity mapping or as more direct connections, such as axosomatic synapses (Figure 1(a)) and electrical synapses (gap junctions) (Fréal et al., 2023; Tewari et al., 2024; Farsang et al., 2024). The dynamics of this model are defined as:

$$
\begin{cases}
v_d[t] = \alpha_1 v_d[t-1] + \beta_1 v_s[t-1] + (1-\alpha_1)c[t] - \gamma_1 s_d[t-1], \\
v_s[t] = \alpha_2 v_s[t-1] + \beta_2 v_d[t] + (1-\alpha_2)c[t] - \gamma_2 s_s[t-1], \\
s_d[t] = H(v_d[t] - v_{\text{th}}), \\
s_s[t] = H(v_s[t] - v_{\text{th}}).
\end{cases}
\tag{5}
$$

In this model, the shortcut is implemented through the term $(1-\alpha_2)c[t]$, allowing direct current input to the soma as shown in Figure 1.

When $\alpha_1$ is close to 1, the dendritic membrane potential $v_d[t]$ acts like a moving average, capturing long-term features by focusing on low-frequency components. Conversely, when $\alpha_2$ is close to 0, the somatic potential $v_s[t]$ rapidly adapts to changes in $c[t]$, making the neuron highly responsive to rapid fluctuations. This dual-compartment model can therefore effectively handle both short-term and long-term signals, providing a balanced approach to temporal processing.

Both dendritic ($s_d$) and somatic ($s_s$) compartments can generate spikes, similar to biological neurons like hippocampal and cortical pyramidal neurons (Muller et al., 2023; Hayashi-Takagi, 2023; Narayanan et al., 2024). The simultaneous firing of multiple types of spikes is also observed in multi-compartment neurons and Hierarchical Temporal Memory (HTM) neurons (Payeur et al., 2021; Capone et al., 2023; Hawkins & Ahmad, 2016). $\gamma_1$ and $\gamma_2$ represent adaptive reset mechanisms rather than fixed voltage resets, providing greater flexibility in the neuron's response. Furthermore, the dendritic and somatic compartments can be modeled as two distinct LIF neurons with unique characteristics, interacting and collaborating to process signals across different frequencies, such as in the visual and auditory systems (Dallos et al., 1972; Wang & Kefalov, 2011). The intensity of their interaction is modulated by the parameter $\beta$ which governs the strength of their mutual influence.

To leverage multi-scale information, we combine dendritic and somatic spike outputs through a weighted sum:

$$
s_{\text{mix}}[t] = \kappa s_d[t] + (1-\kappa)s_s[t],
\tag{6}
$$

where $\kappa$ controls the balance between dendritic and somatic contributions. This coefficient can adapt across different feature channels, allowing flexible integration of both low- and high-frequency components, thereby enabling the model to capture richer temporal features and produce more robust representations of input signals. Here, all the coefficients ($\alpha, \beta, \gamma, \kappa$) mentioned above are learnable.

## 4.2 STABILITY ANALYSIS

To ensure the robustness of the proposed TS-LIF model, we perform a stability analysis by focusing on the homogeneous part of the system (Chen, 1984). The goal is to determine the conditions under which the system remains stable, meaning all solutions remain bounded over time.

**Theorem 1.** *The system governed by the following dynamics:*

$$
\begin{cases}
v_d[t] = \alpha_1 v_d[t-1] + \beta_1 v_s[t-1], \\
v_s[t] = \alpha_2 v_s[t-1] + \beta_2 v_d[t],
\end{cases}
\tag{7}
$$

*has eigenvalues:*

$$
\lambda = \frac{\alpha_1 + \alpha_2 + \beta_1\beta_2 \pm \sqrt{(\alpha_1 + \alpha_2 + \beta_1\beta_2)^2 - 4\alpha_1\alpha_2}}{2}.
\tag{8}
$$

*For the system to remain stable, it is necessary that $|\lambda| < 1$ for both eigenvalues.*

*Proof.* We start by representing the system dynamics in matrix form. The system can be expressed as:

$$\mathbf{v}[t] = A\mathbf{v}[t-1], \tag{9}$$

where the state vector is $\mathbf{v}[t] = \begin{bmatrix} v_d[t] \\ v_s[t] \end{bmatrix}$, and the system matrix is:

$$A = \begin{bmatrix} \alpha_1 & \beta_1 \\ \alpha_1\beta_2 & \alpha_2 + \beta_1\beta_2 \end{bmatrix}. \tag{10}$$

To determine the eigenvalues of the system, we solve the characteristic equation:

$$\det(A - \lambda I) = 0, \tag{11}$$

which results in the quadratic equation:

$$\lambda^2 - (\alpha_1 + \alpha_2 + \beta_1\beta_2)\lambda + \alpha_1\alpha_2 = 0. \tag{12}$$

Solving this quadratic equation gives the eigenvalues:

$$\lambda = \frac{\alpha_1 + \alpha_2 + \beta_1\beta_2 \pm \sqrt{(\alpha_1 + \alpha_2 + \beta_1\beta_2)^2 - 4\alpha_1\alpha_2}}{2}. \tag{13}$$

For stability, both eigenvalues must satisfy $|\lambda| < 1$, ensuring they lie within the unit circle, thereby guaranteeing the boundedness of the system over time. □

The stability of the system is governed by the eigenvalues of matrix $A$. Stability is achieved when both eigenvalues lie within the unit circle, which occurs only if the sum $\alpha_1 + \alpha_2 + \beta_1\beta_2$ is less than 2. This explains why the TC-LIF model, with $\alpha_1 = \alpha_2 = 1$, requires $\beta_1\beta_2 \leq 0$ to ensure stability (Zhang et al., 2024). Properly balancing the interactions between the dendritic and somatic compartments is crucial to maintaining stability.

### 4.3 FREQUENCY RESPONSE ANALYSIS

To examine the frequency characteristics of TS-LIF, we derive the transfer functions by applying the $Z$-transform to the system equations, while ignoring the effects of spike generation and reset mechanisms for simplicity (Chen, 1984). The discrete-time system governing the dynamics of the dendritic and somatic potentials is described by:

$$\begin{cases} v_d[t] = \alpha_1 v_d[t-1] + \beta_1 v_s[t-1] + (1-\alpha_1)c[t], \\ v_s[t] = \alpha_2 v_s[t-1] + \beta_2 v_d[t] + (1-\alpha_2)c[t]. \end{cases} \tag{14}$$

Applying the $Z$-transform under zero initial conditions yields:

$$\begin{cases} (1 - \alpha_1 z^{-1})v_d(z) - \beta_1 z^{-1}v_s(z) = (1-\alpha_1)c(z), \\ -\beta_2 v_d(z) + (1 - \alpha_2 z^{-1})v_s(z) = (1-\alpha_2)c(z). \end{cases} \tag{15}$$

The system can then be expressed as:

$$\mathbf{M}\mathbf{v}(z) = \tilde{\mathbf{c}}(z). \tag{16}$$

where

$$\mathbf{M} = \begin{bmatrix} 1 - \alpha_1 z^{-1} & -\beta_1 z^{-1} \\ -\beta_2 & 1 - \alpha_2 z^{-1} \end{bmatrix}, \quad \mathbf{v}(z) = \begin{bmatrix} v_d(z) \\ v_s(z) \end{bmatrix}, \quad \tilde{\mathbf{c}}(z) = \begin{bmatrix} (1-\alpha_1)c(z) \\ (1-\alpha_2)c(z) \end{bmatrix}.$$

The transfer functions for $v_d[t]$ and $v_s[t]$ with respect to the input $c[t]$ are:

$$H_d(z) = \frac{v_d(z)}{c(z)} = \frac{(1-\alpha_1)(1 - \alpha_2 z^{-1}) + \beta_1 z^{-1}(1-\alpha_2)}{\det(\mathbf{M})},$$

$$H_s(z) = \frac{v_s(z)}{c(z)} = \frac{\beta_2(1-\alpha_1) + (1 - \alpha_1 z^{-1})(1-\alpha_2)}{\det(\mathbf{M})}, \tag{17}$$

where the determinant of $\mathbf{M}$ is:

$$\det(\mathbf{M}) = 1 - (\alpha_1 + \alpha_2 + \beta_1\beta_2)z^{-1} + \alpha_1\alpha_2 z^{-2}. \tag{18}$$

These transfer functions offer insights into how the dendritic and somatic compartments process different frequency components of the input signal. The magnitudes of $H_d(z)$ and $H_s(z)$ represent the system's gain at specific frequencies, while their phases indicate the delay or shift introduced by the system.

### 4.4 EXAMPLE: FREQUENCY SEPARATION IN PRACTICE

To illustrate the system's behavior, we use the following parameter settings: $\alpha_1 = 0.95$, which enables low-pass filtering in $v_d[t]$, $\alpha_2 = 0.05$, providing high-pass filtering for $v_s[t]$, along with $\beta_1 = 0$ and $\beta_2 = -0.9$. Under these conditions, the characteristic equation for the system is derived as:

$$1 - \lambda + 0.0475\lambda^2 = 0, \tag{19}$$

Solving this, we find the eigenvalues $\lambda_1 = 0.95$ and $\lambda_2 = 0.05$, both of which lie within the unit circle, ensuring system stability. The corresponding transfer functions for the dendritic and somatic compartments are:

$$H_d(z) = \frac{0.05 - 0.0025z^{-1}}{1 - z^{-1} + 0.0475z^{-2}}, \tag{20}$$

$$H_s(z) = \frac{0.905 - 0.9025z^{-1}}{1 - z^{-1} + 0.0475z^{-2}}. \tag{21}$$

At low frequencies ($\omega = 0$), $H_d(e^{j\omega}) = 1$, indicating that $v_d[t]$ effectively captures low-frequency signals, while $H_s(e^{j\omega}) \approx 0.0526$, showing minimal response. Conversely, at high frequencies ($\omega = \pi$), $H_s(e^{j\omega}) \approx 0.8829$, demonstrating that $v_s[t]$ accurately reflects high-frequency components. This example highlights the system's frequency separation properties, where $v_d[t]$ captures slow-changing signals, and $v_s[t]$ is sensitive to rapid changes.

## 5 EXPERIMENTS

In this section, we present the experimental evaluation of the TS-LIF model across multiple time-series benchmarks. We analyze the model's forecasting performance, its robustness against various prediction settings, and its ability in temporal decomposition.

### 5.1 TEMPORAL ANALYSIS

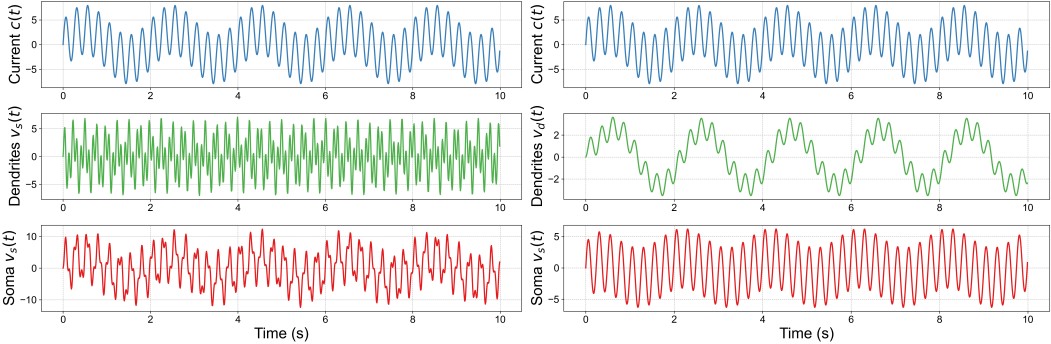

Figure 2: (a) TC-LIF Model Response          Figure 3: (b) TS-LIF Model Response

Figure 4: Comparison of dendritic and somatic voltage responses between TC-LIF and TS-LIF models with mixed-frequency input current.

We performed a temporal analysis to assess the TS-LIF model's ability to decompose input signals into distinct frequency components. To validate this temporal decomposition capability, we fed a mixed-frequency input current (blue line) into both TS-LIF and TC-LIF models, as depicted by the blue line in Figure 4. Detailed experimental settings are provided in Appendix A.4.

By visualizing the voltage responses of the dendritic (green line) and somatic (red line) compartments, we observed distinct behaviors between the models. The TS-LIF model effectively separated

the input signal into its low- and high-frequency components, with the dendritic compartment retaining primarily low-frequency information and the somatic compartment focusing on high-frequency components. In contrast, the TC-LIF model did not demonstrate a clear separation, with both compartments reflecting mixed-frequency components. These results emphasize the advantage of the TS-LIF model in managing temporal information, effectively capturing and processing multi-timescale dependencies in time series — an essential capability for handling complex temporal patterns.

## 5.2 Main Results

Table 1: Forecasting results on four benchmark datasets with different prediction horizons $L$. Results for our TS-LIF model with TCN, GRU, and Transformer architectures are included, while the remaining results are sourced from Lv et al. (2024). The top-performing and second-best scores are shown in bold and underlined, respectively. Arrows ↑ (↓) denote whether higher or lower values are preferred. The **Avg. Rank** column reflects the average rank of each model across the different configurations.

| Method | Spike | Metric | Metr-la | | | | Pems-bay | | | | Solar | | | | Electricity | | | | Avg. | Avg. Rank↓ |
|---|---|---|---|---|---|---|---|---|---|---|---|---|---|---|---|---|---|---|---|---|
| | | | 6 | 24 | 48 | 96 | 6 | 24 | 48 | 96 | 6 | 24 | 48 | 96 | 6 | 24 | 48 | 96 | | |
| ARIMA | ✗ | $R^2$↑ | .687 | .441 | .282 | .265 | .741 | .723 | .692 | .670 | .951 | .847 | .725 | .682 | .963 | .960 | .914 | .863 | .713 | 9.9 |
| | | RSE↓ | .575 | .742 | .889 | .902 | .532 | .548 | .562 | .612 | .202 | .365 | .588 | .589 | .522 | .534 | .564 | .599 | .583 | 9.8 |
| GP | ✗ | $R^2$↑ | .685 | .437 | .265 | .233 | .732 | .712 | .689 | .665 | .944 | .836 | .711 | .675 | .962 | .968 | .912 | .852 | .705 | 11.1 |
| | | RSE↓ | .572 | .738 | .912 | .925 | .544 | .532 | .577 | .592 | .225 | .388 | .612 | .575 | .603 | .612 | .633 | .642 | .605 | 10.2 |
| TCN | ✗ | $R^2$↑ | .820 | .601 | .455 | **.330** | .881 | .749 | .695 | **.689** | .958 | .871 | .737 | .661 | .975 | .973 | .968 | .962 | .770 | 5.8 |
| | | RSE↓ | .446 | .665 | .778 | **.851** | .373 | .541 | .583 | .587 | .210 | .359 | .513 | .583 | .282 | .287 | **.319** | .345 | .483 | 5.7 |
| Spike-TCN | ✓ | $R^2$↑ | .783 | .603 | **.468** | .326 | .811 | .729 | .662 | .633 | .937 | .840 | .708 | .650 | .970 | .963 | .958 | .953 | .750 | 7.0 |
| | | RSE↓ | .491 | .665 | .769 | .865 | .469 | .541 | .625 | .635 | .259 | .401 | .541 | .596 | .333 | .342 | .368 | .389 | .518 | 8.9 |
| **TS-TCN** | ✓ | $R^2$↑ | .810 | .605 | **.473** | .328 | **.897** | **.759** | **.698** | .652 | **.964** | **.884** | .762 | .720 | .980 | .971 | .968 | .962 | .777 | 5.0 |
| | | RSE↓ | .459 | .656 | .757 | .857 | **.354** | **.527** | .590 | .633 | **.189** | **.325** | .484 | .523 | .264 | .316 | .318 | .360 | .475 | 4.6 |
| GRU | ✗ | $R^2$↑ | .759 | .429 | .301 | .194 | .747 | .703 | .691 | .665 | .950 | .875 | .781 | .737 | .981 | .972 | .971 | .964 | .733 | 8.0 |
| | | RSE↓ | .517 | .797 | .882 | .947 | .529 | .573 | .584 | .608 | .219 | .355 | .476 | .522 | .506 | .598 | .537 | .587 | .573 | 9.5 |
| Spike-GRU | ✓ | $R^2$↑ | .846 | .615 | .427 | .275 | .864 | .741 | .688 | .657 | .912 | .822 | .771 | .668 | .978 | .964 | .962 | .959 | .759 | 8.8 |
| | | RSE↓ | .414 | .663 | .827 | .943 | .398 | .535 | .601 | .621 | .299 | .430 | .485 | .629 | .280 | .317 | .338 | .484 | .517 | 8.8 |
| Spike-RNN | ✓ | $R^2$↑ | .846 | .622 | .433 | .283 | .872 | .745 | .685 | .654 | .923 | .820 | .812 | .714 | .977 | .972 | .962 | .960 | .768 | 7.4 |
| | | RSE↓ | .412 | .648 | .794 | .935 | .387 | .528 | .588 | .634 | .278 | .425 | .435 | .586 | .267 | .296 | .346 | .481 | .503 | 7.1 |
| **TS-GRU** | ✓ | $R^2$↑ | **.848** | .618 | .430 | .329 | .874 | .742 | .684 | .649 | .938 | .878 | .815 | .722 | **.991** | .981 | **.983** | .976 | .778 | 4.8 |
| | | RSE↓ | **.412** | .651 | .795 | .853 | .384 | .530 | .587 | .637 | .253 | .349 | .426 | .527 | .216 | .240 | .236 | .271 | .460 | 4.8 |
| Autoformer | ✗ | $R^2$↑ | .762 | .548 | .411 | .282 | .782 | .711 | .689 | .668 | .960 | .852 | .791 | .701 | .980 | .977 | .975 | .963 | .753 | 7.2 |
| | | RSE↓ | .565 | .692 | .785 | .872 | .452 | .543 | .577 | .565 | .212 | .432 | .622 | .685 | .481 | .506 | .566 | .548 | .569 | 9.1 |
| iTransformer | ✗ | $R^2$↑ | .829 | **.623** | .439 | .285 | .887 | .719 | .685 | .668 | **.964** | .879 | .799 | .738 | .979 | .977 | .975 | .964 | .776 | 4.4 |
| | | RSE↓ | .436 | **.648** | .780 | .878 | .362 | .547 | **.561** | .584 | **.191** | .348 | .448 | .563 | .259 | .305 | .335 | .427 | .480 | 4.7 |
| iSpikformer | ✓ | $R^2$↑ | .817 | .618 | .440 | .279 | .879 | .744 | .687 | .674 | .961 | .876 | .795 | .738 | .977 | .974 | .972 | .963 | .775 | 5.2 |
| | | RSE↓ | .475 | .668 | .752 | .905 | .376 | .536 | .569 | .580 | .204 | .333 | .465 | .521 | .263 | .284 | .338 | .348 | .476 | 4.6 |
| **TS-former** | ✓ | $R^2$↑ | .847 | .620 | .445 | .283 | .874 | .735 | .683 | .669 | .961 | **.886** | **.828** | **.774** | .987 | **.985** | .981 | **.977** | **.783** | **3.5** |
| | | RSE↓ | .416 | .655 | .763 | .874 | .379 | .539 | .572 | .583 | .224 | .331 | **.382** | **.435** | **.197** | **.215** | **.234** | **.261** | **.441** | **3.3** |

The proposed TS-LIF model (TS-TCN, TS-GRU, and TS-former) was evaluated on four benchmark datasets (Metr-la, Pems-bay, Solar, and Electricity), following the setup in Lv et al. (2024), where TS-former represents an iTransformer architecture based on TS-LIF (Liu et al., 2024). Hyperparameters were tuned via cross-validation, and performance was assessed using RSE and $R^2$ metrics. As shown in Table 1, Our TS-LIF consistently outperforms LIF-based SNN models (Spike-TCN, Spike-GRU, and iSpikeformer) across different metrics, particularly excelling in tasks requiring long-term forecasting. For example, in the Solar dataset, the TS-TCN and TS-former achieved an average improvement of 8% in $R^2$ and 16.8% in RSE. Similarly, in the Electricity dataset, TS-GRU and TS-former showed significant improvements in RSE of 43.6% and 25.0%, respectively, for the 96-step prediction length. These results highlight the effectiveness of TS-LIF in capturing long-term dependencies, unlike traditional LIF neurons.

For RNN-based models, our TS-LIF achieved superior performance in both $R^2$ and RSE metrics compared to LIF-based and original ANN models. Even for TCN and Transformer models, which inherently possess cross-time-scale capabilities, TS-LIF aslo provided noticeable improvements, resulting in higher average rankings across metrics. This suggests that the integration of dendritic and somatic components in the TS-LIF framework enables the model to capture richer multi-scale temporal features, leading to improved predictive accuracy.

## 5.3 Model Analysis

This section assesses model robustness in the presence of missing values, evaluates the impact of training timesteps, and examines the effectiveness of different types of LIF neurons.

Table 2: Experimental performance of the TS-LIF model compared to the vanilla LIF on the Electricity dataset, evaluated under different ratios of missing values in the historical inputs. Model_* indicates a backbone model with a prediction length of *, and Transformer_6 represents the Transformer architecture with a prediction length of 6.

| Missing Ratio | | 10% | | 20% | | 40% | | 60% | | 80% | |
|---|---|---|---|---|---|---|---|---|---|---|---|
| Metric | | $R^2\uparrow$ | RSE↓ | $R^2\uparrow$ | RSE↓ | $R^2\uparrow$ | RSE↓ | $R^2\uparrow$ | RSE↓ | $R^2\uparrow$ | RSE↓ |
| Transformer_6 | iSpikformer | 0.977 | 0.265 | 0.976 | 0.266 | 0.974 | 0.269 | 0.971 | 0.271 | 0.964 | 0.275 |
| | **TS-former** | **0.987** | **0.199** | **0.987** | **0.200** | **0.987** | **0.206** | **0.986** | **0.207** | **0.983** | **0.211** |
| | Promotion | 1.0% | 24.9% | 1.1% | 24.8% | 1.3% | 23.4% | 1.5% | 23.6% | 1.9% | 23.2% |
| Transformer_96 | iSpikformer | 0.962 | 0.344 | 0.962 | 0.346 | 0.957 | 0.358 | 0.953 | 0.368 | 0.947 | 0.376 |
| | **TS-former** | **0.977** | **0.263** | **0.976** | **0.262** | **0.974** | **0.270** | **0.971** | **0.274** | **0.971** | **0.279** |
| | Promotion | 1.5% | 23.5% | 1.4% | 24.2% | 1.7% | 24.5% | 1.8% | 25.5% | 2.4% | 25.7% |
| GRU_6 | Spike-GRU | 0.873 | 0.774 | 0.842 | 0.749 | 0.771 | 0.851 | 0.758 | 0.874 | 0.745 | 0.897 |
| | **TS-GRU** | **0.986** | **0.235** | **0.983** | **0.242** | **0.979** | **0.256** | **0.965** | **0.328** | **0.939** | **0.438** |
| | Promotion | 12.9% | 69.6% | 16.7% | 67.6% | 26.9% | 69.9% | 27.3% | 62.4% | 26.0% | 51.1% |
| GRU_96 | Spike-GRU | 0.842 | 0.693 | 0.827 | 0.729 | 0.803 | 0.789 | 0.783 | 0.828 | 0.760 | 0.871 |
| | **TS-GRU** | **0.975** | **0.268** | **0.973** | **0.324** | **0.970** | **0.303** | **0.956** | **0.367** | **0.922** | **0.489** |
| | Promotion | 15.7% | 61.3% | 17.6% | 55.6% | 20.7% | 61.5% | 22.0% | 55.6% | 21.3% | 43.8% |
| TCN_6 | Spike-TCN | 0.971 | 0.341 | 0.970 | 0.347 | 0.967 | 0.352 | 0.960 | 0.361 | 0.954 | 0.369 |
| | **TS-TCN** | **0.980** | **0.263** | **0.979** | **0.266** | **0.976** | **0.272** | **0.973** | **0.280** | **0.967** | **0.291** |
| | Promotion | 0.9% | 22.8% | 1.0% | 23.3% | 1.0% | 22.7% | 0.7% | 22.4% | 1.3% | 21.1% |
| TCN_96 | Spike-TCN | 0.953 | 0.390 | 0.952 | 0.394 | 0.948 | 0.409 | 0.942 | 0.418 | 0.937 | 0.426 |
| | **TS-TCN** | **0.962** | **0.361** | **0.961** | **0.364** | **0.958** | **0.370** | **0.956** | **0.375** | **0.952** | **0.389** |
| | Promotion | 1.0% | 7.4% | 0.9% | 7.6% | 1.0% | 9.5% | 1.4% | 10.2% | 1.6% | 8.6% |

### 5.3.1 ROBUSTNESS EVALUATION

To verify the robustness of the proposed TS-LIF model, we evaluated its performance on the Electricity dataset under different ratios of missing values in the historical inputs, comparing it to the vanilla LIF-based models. The models were assessed under missing data ratios of 10%, 20%, 40%, 60%, and 80%. The results are presented in Table 2.

The experimental results indicate that TS-LIF consistently outperforms the vanilla LIF models across different missing value scenarios, as evidenced by higher $R^2$ values and lower RSE scores. Compared to the LIF-based models, TS-LIF shows a significantly smaller reduction in prediction accuracy as the missing data ratio increases, particularly in GRU-based models. Since GRU inherently has a weaker capability to capture long-term dynamics compared to TCN and Transformer, the enhancements introduced by TS-LIF greatly improve its temporal feature extraction ability. Specifically, TS-LIF improves $R^2$ by approximately 20% and reduces RSE by around 50% compared to the baseline LIF-based models. These improvements highlight the effectiveness of TS-LIF in capturing complex temporal dependencies and maintaining robustness under challenging conditions with substantial data loss.

### 5.3.2 TRAINING TIMESTEPS

Figure 5 illustrates the forecasting performance of our TS-LIF model, evaluated using TCN, GRU, and Transformer architectures over different training timesteps (24, 96, and 168) for the Solar and Electricity datasets. For each architecture, the TS-LIF model consistently improves performance as the training timesteps increase from 24 to 168. This trend is evident from the rising $R^2$ values and decreasing RSE scores, indicating enhanced accuracy with more extended training. Notably, the most substantial improvements are observed in the transition from 24 to 96 timesteps, showcasing the model's capability to leverage longer training periods to capture complex temporal patterns more effectively.

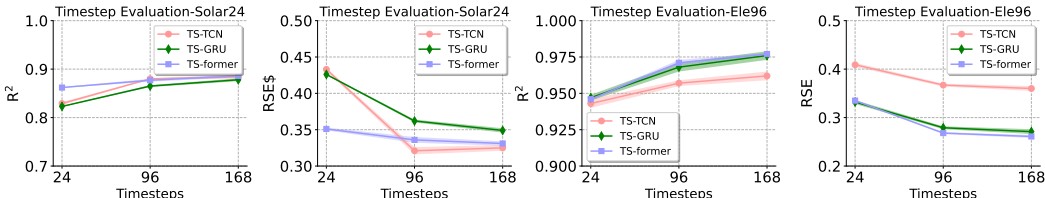

Figure 5: Impact of Training Timesteps on Forecasting Performance of TS-LIF Model Across Different Architectures. Solar dataset with a prediction length of 24 and Electricity dataset with a prediction length of 96. We plot mean and std for each experiment over 3 different random seeds.

### 5.3.3 LIF Neurons

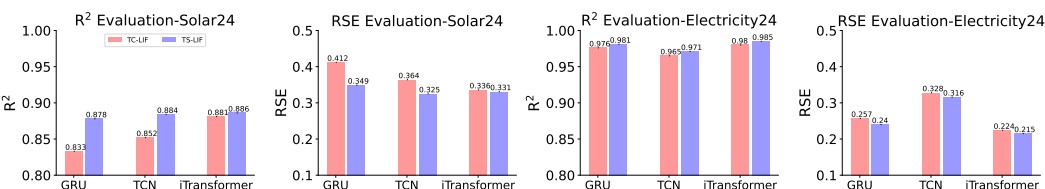

Figure 6: Forecasting Accuracy Comparison of TS-LIF and TC-LIF Neurons on Solar and Electricity datasets with a prediction length of 24.

In this section, we evaluate the accuracy of the proposed TS-LIF model compared to the dual-compartment TC-LIF model. As shown in Figure 6, TS-LIF consistently outperforms TC-LIF across all three architectures and both datasets. The $R^2$ values for TS-LIF are significantly higher, particularly with GRU and TCN, demonstrating improved predictive accuracy. The corresponding RSE scores also confirm this trend, with TS-LIF showing consistently lower errors, indicating better fit and reduced prediction errors. These improvements highlight TS-LIF's superior capability to capture underlying temporal patterns and effectively manage multi-scale information in the datasets.

## 6 Conclusion

In this work, we introduced the Temporal Segment Leaky Integrate-and-Fire (TS-LIF) model, a novel spiking neural network (SNN) neuron architecture designed specifically for time series forecasting. The TS-LIF model features a dual-compartment structure, with dendritic and somatic compartments processing different frequency components, allowing for effective multi-timescale information integration. This compartmentalization enables TS-LIF to enhance both low- and high-frequency signal processing, addressing key limitations of traditional Leaky Integrate-and-Fire (LIF) neurons in capturing long-term dependencies and multi-scale dynamics.

We theoretically proved the stability conditions for the TS-LIF model, ensuring robustness across a wide range of temporal inputs. Through frequency response analysis, we demonstrated how the dendritic and somatic compartments contribute to efficient temporal decomposition. Our empirical evaluation on four benchmark datasets showed that the TS-LIF model consistently outperformed conventional LIF-based SNNs as well as artificial neural networks, particularly in long-term forecasting scenarios. Moreover, the TS-LIF model showed resilience under missing input data, maintaining superior accuracy compared to baseline models. The proposed model advances the state-of-the-art in SNNs for time series forecasting by combining biologically inspired design with computational efficiency, providing a promising solution for applications requiring robust temporal processing in resource-constrained environments.

## 7 ACKNOWLEDGMENT

This research is supported by the Joint NTU-WeBank Research Centre on Fintech, Nanyang Technological University, Singapore.

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

# A APPENDIX

## A.1 EXPERIMENT SETTINGS FOR MAIN RESULTS

In this section, we outline the experimental setup used to evaluate the performance of the proposed TD-LIF model. We conducted experiments on several benchmark time series datasets, including Metr-LA (Li et al., 2017), which records average traffic speed on highways in Los Angeles County; Pems-Bay (Li et al., 2017), capturing traffic speed data in the Bay Area; Electricity (Lai et al., 2018), which tracks hourly electricity consumption in kWh; and Solar (Lai et al., 2018), detailing solar power production. Preprocessing steps were applied to ensure consistency across datasets, standardizing input dimensions, sampling rates, and data normalization.

The TS-LIF model framework was implemented in line with the approach in Lv et al. (2024), incorporating CNN-based TCNs (Bai et al., 2018), RNN-based GRUs (Cho, 2014), and Transformer-based models such as Autoformer (Wu et al., 2021) and iTransformer (Liu et al., 2024). As for the SNN-based structure, we introduce the TCN, RNN, GRUs, and Transformer models of the SNN format from Lv et al. (2024). Hyperparameters, including learning rate, timestep intervals, and feature-mixing weights, were optimized through cross-validation. We employ two statistical metrics: the Root Relative Squared Error (RSE) and the coefficient of determination ($R^2$) followed by the Lv et al. (2024) settings. Detailed descriptions of the experimental settings and hyperparameter configurations are provided in the appendix.

## A.2 DATASET AND METRIC DETAILS

**Datasets**. The details of the datasets used in the main experiment are shown in Table 3. In the experimental partitioning of datasets Metr-la and Pems-bay, we adopted a train-validation-test ratio of 0.7, 0.2, and 0.1, respectively, while for datasets Solar and Electricity, we used ratios of 0.6, 0.2, and 0.2. The settings for history and prediction lengths in the experiments followed those in the paper by Lv et al. (2024), except that for the history length in datasets Metr-la and Pems-bay, we added a setting of 168 to further improve experimental performance.

Table 3: Properties of the datasets in experiments

| DATASET | Dimension | Domain | Freq | Samples | Context Length | Pred Length |
|---------|-----------|--------|------|---------|----------------|-------------|
| Metr-la | 207 | $\mathbb{R}^+$ | 30-min | 34,272 | $\{12, 168\}$ | $\{6, 24, 48, 96\}$ |
| Pems-bay | 325 | $\mathbb{R}^+$ | 30-min | 52,116 | $\{12, 168\}$ | $\{6, 24, 48, 96\}$ |
| Solar | 137 | $\mathbb{R}^+$ | Hourly | 52,560 | 168 | $\{6, 24, 48, 96\}$ |
| Electricity | 321 | $\mathbb{R}^+$ | Hourly | 26,304 | 168 | $\{6, 24, 48, 96\}$ |

**Metrics**. To comprehensively evaluate our model's performance, we employ two statistical metrics: the Root Relative Squared Error (RSE) and the coefficient of determination ($R^2$). The RSE measures the relative discrepancy between the predicted and actual values, while the $R^2$ indicates the proportion of variance in the dependent variable that is predictable from the independent variables. These metrics are calculated as follows:

$$
\text{RSE} = \sqrt{\frac{\sum_{m=1}^{M} ||\mathbf{Y}^m - \hat{\mathbf{Y}}^m||^2}{\sum_{m=1}^{M} ||\mathbf{Y}^m - \bar{\mathbf{Y}}||^2}},
$$

$$
R^2 = \frac{1}{MCL} \sum_{m=1}^{M} \sum_{c=1}^{C} \sum_{l=1}^{L} \left[ 1 - \frac{(Y_{c,l}^m - \hat{Y}_{c,l}^m)^2}{(Y_{c,l}^m - \bar{Y}_{c,l})^2} \right],
$$

$$(22)$$

where $M$ denotes the number of samples in the test set, $C$ represents the number of channels or variables, and $L$ is the prediction horizon. The true values for the $m$-th sample are denoted by $\mathbf{Y}^m$, and their average over all samples is $\bar{\mathbf{Y}}$. Specifically, $Y_{c,l}^m$ represents the $l$-th future value of the $c$-th variable for the $m$-th sample, with its mean across all samples given by $\bar{Y}_{c,l}$. The predicted values corresponding to these true values are denoted by $\hat{\mathbf{Y}}^m$ and $\hat{Y}_{c,l}^m$, respectively.

## A.3 IMPLEMENTATION DETAILS

In this section, we summarize the detailed experiment setup of our TS-LIF. Table 4 and 5 show the hyperparameters of our overall structure in three types of backbones (TCN, GRU, Transformer). As for the timesteps in the SNN structures, we align them with the history length in each setting. The threshold of the TS-LIF is set to 1.0.

Table 4: Hyperparameters of different backbones (TCN, GRU, Transformer) used for each dataset

| Datesets | TCN Layers | TCN Kernels | GRU Layers | Transformer Layers | Attention Heads | Attention Dim |
|---|---|---|---|---|---|---|
| Metr-la | 3 | 3 | 1 | 2 | 8 | 256 |
| Pems-bay | 3 | 16 | 1 | 2 | 8 | 512 |
| Solar | 3 | 16 | 1 | 2 | 8 | 512 |
| Electricity | 3 | 3 | 1 | 2 | 8 | 256 |

Table 5: Training details of different backbones (TCN, GRU, Transformer) used for each dataset

| Datesets | TCN Hidden | TCN Dilation | GRU Hidden | Transformer d_ff | Learning Rate | Batch Size |
|---|---|---|---|---|---|---|
| Metr-la | 64 | 2 | 128 | 1024 | {.0001, .0005} | {32, 64 } |
| Pems-bay | 64 | 2 | 128 | 2048 | {.0001, .0005} | {32, 64 } |
| Solar | 64 | 2 | 128 | 2048 | .0001 | 64 |
| Electricity | 64 | 2 | 128 | 1024 | .0001 | 64 |

## A.4 EXPERIMENT SETTINGS FOR TEMPORAL ANALYSIS

The injected current, $I(t)$, consists of two sinusoidal components: a low-frequency component, $I_{\text{low\_freq}}$, with an amplitude of 3 and a frequency of 0.5 Hz, and a high-frequency component, $I_{\text{high\_freq}}$, with an amplitude of 5 and a frequency of 4 Hz. This combination represents a complex input environment with both slow and rapid variations, simulating mixed-frequency stimuli.

For the TCLIF model, we adopted parameters $\alpha_1 = \alpha_2 = 1$, $\beta_1 = -0.5$, and $\beta_2 = 0.5$, as suggested in Zhang et al. (2024). In contrast, the TS-LIF model was set with $\alpha_1 = 0.95$, $\alpha_2 = 0.05$, $\beta_1 = 0$, and $\beta_2 = -0.9$, which corresponds to the parameter settings used in the frequency response analysis in the previous subsection. These values were selected to facilitate low-pass filtering in the dendritic compartment ($v_d[t]$) and high-pass filtering in the somatic compartment ($v_s[t]$).

## A.5 THEORETICAL ENERGY CONSUMPTION CALCULATION

The theoretical energy consumption for each layer during inference is determined based on the operations performed by spiking neural networks (SNNs) and artificial neural networks (ANNs) (Yao et al., 2023).

For SNNs, the energy required by layer $l$ is calculated as:

$$Energy(l) = E_{AC} \times SOPs(l),$$

where $SOPs(l)$ is the number of spike-based accumulate (AC) operations, and $E_{AC}$ represents the energy per AC operation.

For ANNs, the energy consumption for layer $b$ is:

$$Energy(b) = E_{MAC} \times FLOPs(b),$$

where $FLOPs(b)$ refers to the number of floating-point multiply-and-accumulate (MAC) operations, and $E_{MAC}$ is the energy per MAC operation. The constants are set as $E_{MAC} = 4.6 \, \text{pJ}$ and $E_{AC} = 0.9 \, \text{pJ}$, assuming operations are performed on 45nm hardware.

For SNNs, the number of synaptic operations in layer $l$ is further estimated as:

$$SOPs(l) = T \times \gamma \times FLOPs(l),$$

where $T$ is the number of timesteps required in the simulation, and $\gamma$ is the firing rate of the input spike train for layer $l$.

Table 6: Energy consumption per sample of the Electricity dataset during inference. "OPs" includes SOPs for SNNs and FLOPs for ANNs. "SOPs" refers to synaptic operations in SNNs, and "FLOPs" denotes floating-point operations in ANNs.

| Model | Param(M) | OPs (G) | Energy (mJ) | Energy Reduction | Train/Infer Time (s) | $R^2$ |
|---|---|---|---|---|---|---|
| TCN | 0.460 | 0.14 | 0.64 | - | 21.34/11.47 | .973 |
| Spike-TCN | 0.461 | 0.15 | 0.23 | 63.60% ↓ | 306.91/27.85 | .963 |
| **TS-TCN** | 0.465 | 0.19 | 0.25 | 60.93% ↓ | 308.26/28.14 | .971 |
| GRU | 1.288 | 1.32 | 6.07 | - | 37.73/7.35 | .972 |
| Spike-GRU | 1.289 | 1.63 | 1.51 | 75.05% ↓ | 235.46/10.05 | .964 |
| **TS-GRU** | 1.291 | 1.67 | 1.58 | 73.80% ↓ | 246.23/9.78 | **.981** |
| iTransformer | 1.634 | 2.05 | 9.47 | - | 7.24/6.38 | .977 |
| iSpikformer | 1.634 | 3.55 | 3.19 | 66.30% ↓ | 49.84/8.69 | .974 |
| **TS-former** | 1.640 | 3.59 | 3.22 | 65.99% ↓ | 50.36/8.72 | **.985** |

Table 7: Performance of our TS-former with SparseTSF and SAMformer of 3 prediction lengths (24, 48, 96) on the Metr-la and Electricity datasets. SparseTSF*: replace the ReLU function of SparseTSF with our TS-LIF. **Bold** numbers represent the best outcomes.

| Datasets | Metr-la | | | | | | Electricity | | | | | |
|---|---|---|---|---|---|---|---|---|---|---|---|---|
| Lengths | 24 | | 48 | | 96 | | 24 | | 48 | | 96 | |
| Metrics | $R^2$↑ | RSE↓ | $R^2$↑ | RSE↓ | $R^2$↑ | RSE↓ | $R^2$↑ | RSE↓ | $R^2$↑ | RSE↓ | $R^2$↑ | RSE↓ |
| SparseTSF | 0.576 | 0.681 | 0.427 | 0.792 | 0.253 | 0.916 | **0.991** | **0.167** | **0.986** | **0.195** | **0.982** | **0.232** |
| SparseTSF* | 0.580 | 0.692 | 0.426 | 0.801 | 0.247 | 0.924 | 0.990 | 0.177 | 0.985 | 0.201 | **0.982** | 0.234 |
| SAMformer | 0.549 | 0.739 | 0.401 | 0.863 | 0.219 | 0.965 | 0.983 | 0.218 | 0.980 | 0.239 | 0.978 | 0.257 |
| **TS-former** | **0.620** | **0.655** | **0.445** | **0.763** | **0.283** | **0.874** | 0.985 | 0.215 | 0.981 | 0.234 | 0.977 | 0.261 |

## A.6 PERFORMANCE COMPARISON ON SOTA TIME SERIES FORECASTING METHODS

Table 7 compares the performance of our TS-former with SparseTSF (Lin et al., 2024), SparseTSF* (where ReLU is replaced by TS-LIF), and SAMformer (Ilbert et al., 2024a) on the Metr-la and Electricity datasets for prediction lengths of 24, 48, and 96.

On the Metr-la dataset, TS-former achieves the best results across all metrics and prediction lengths, demonstrating its ability to effectively capture complex temporal dependencies. For example, at a prediction length of 24, our TS-former achieves an $R^2$ of 0.620 and RSE of 0.655, outperforming both SparseTSF and SAMformer. On the Electricity dataset, SparseTSF achieves slightly better performance in some cases, such as an $R^2$ of 0.991 and RSE of 0.167 at a prediction length of 24. However, TS-former remains competitive, delivering consistent and robust results across different prediction lengths. These results highlight the effectiveness of TS-LIF in SparseTSF* and the overall robustness of TS-former in time series forecasting tasks.

## A.7 COMPARISON OF TS-LIF WITH OTHER LIF NEURONS

Table 8 compares the performance of our TS-LIF with TC-LIF, LM-H, and CLIF in the GRU backbone on the Metr-la and Electricity datasets for prediction lengths of 6, 24, and 96. TS-LIF consistently outperforms the baseline methods across all metrics and prediction lengths. For example, on the Metr-la dataset, TS-LIF achieves the highest $R^2$ of 0.848 and the lowest RSE of 0.412 at a prediction length of 6. Similarly, on the Electricity dataset, TS-LIF achieves an $R^2$ of 0.991 and RSE of 0.216 at the same prediction length, demonstrating its robustness and effectiveness in modeling temporal dependencies. These results highlight the superiority of TS-LIF over existing LIF structures, making it a strong choice for time series forecasting tasks.

## A.8 ROBUSTNESS ANALYSIS ON THE METR-LA DATASET

To further verify the robustness of the proposed TS-LIF model, we evaluated its performance on the Metr-la dataset under different ratios of missing values in the historical inputs, comparing it to the

Table 8: Performance of our TS-LIF with other LIF neurons (TC-LIF, LM-H, and CLIF) in the GRU backbone. **Bold** numbers represent the best outcomes.

| Datasets | Metr-la | | | | | | Electricity | | | | | |
|---|---|---|---|---|---|---|---|---|---|---|---|---|
| Lengths | 6 | | 24 | | 96 | | 6 | | 24 | | 96 | |
| Metrics | $R^2\uparrow$ | RSE$\downarrow$ | $R^2\uparrow$ | RSE$\downarrow$ | $R^2\uparrow$ | RSE$\downarrow$ | $R^2\uparrow$ | RSE$\downarrow$ | $R^2\uparrow$ | RSE$\downarrow$ | $R^2\uparrow$ | RSE$\downarrow$ |
| TC-LIF | 0.828 | 0.453 | 0.594 | 0.673 | 0.259 | 0.956 | 0.978 | 0.263 | 0.976 | 0.257 | 0.941 | 0.503 |
| LM-H | 0.812 | 0.464 | 0.570 | 0.719 | 0.246 | 0.973 | 0.971 | 0.269 | 0.969 | 0.280 | 0.936 | 0.512 |
| CLIF | 0.837 | 0.429 | 0.606 | 0.667 | 0.271 | 0.930 | 0.973 | 0.259 | 0.972 | 0.276 | 0.954 | 0.376 |
| **TS-LIF** | **0.848** | **0.412** | **0.618** | **0.651** | **0.329** | **0.853** | **0.991** | **0.216** | **0.981** | **0.240** | **0.976** | **0.271** |

vanilla LIF-based models. The models were assessed under missing data ratios of 10%, 20%, 40%, 60%, and 80%. The results are presented in Table 9.

Table 9: Experimental performance of the TS-LIF model compared to the vanilla LIF on the Metr-la dataset, evaluated under different ratios of missing values in the historical inputs. Model_* indicates a backbone model with a prediction length of *, and Transformer_6 represents the Transformer architecture with a prediction length of 6.

| Missing Ratio | | 10% | | 20% | | 40% | | 60% | | 80% | |
|---|---|---|---|---|---|---|---|---|---|---|---|---|
| Metric | | $R^2\uparrow$ | RSE$\downarrow$ | $R^2\uparrow$ | RSE$\downarrow$ | $R^2\uparrow$ | RSE$\downarrow$ | $R^2\uparrow$ | RSE$\downarrow$ | $R^2\uparrow$ | RSE$\downarrow$ |
| Transformer_6 | iSpikformer | 0.815 | 0.479 | 0.813 | 0.486 | 0.809 | 0.488 | 0.804 | 0.492 | 0.802 | 0.496 |
| | **TS-former** | **0.843** | **0.419** | **0.842** | **0.421** | **0.838** | **0.430** | **0.835** | **0.436** | **0.831** | **0.440** |
| | Promotion | 3.43% | 14.3% | 3.56% | 15.4% | 3.58% | 13.4% | 3.86% | 12.8% | 3.61% | 12.7% |
| Transformer_96 | iSpikformer | 0.270 | 0.915 | 0.254 | 0.926 | 0.230 | 0.935 | 0.204 | 0.948 | 0.194 | 0.959 |
| | **TS-former** | **0.277** | **0.907** | **0.263** | **0.911** | **0.238** | **0.92 7** | **0.212** | **0.936** | **0.205** | **0.945** |
| | Promotion | 2.59% | 0.88% | 3.54% | 1.65% | 3.47% | 0.86% | 3.92% | 1.28% | 5.67% | 1.48% |
| GRU_6 | Spike-GRU | 0.830 | 0.429 | 0.819 | 0.440 | 0.771 | 0.497 | 0.746 | 0.522 | 0.743 | 0.530 |
| | **TS-GRU** | **0.843** | **0.417** | **0.839** | **0.414** | **0.834** | **0.425** | **0.823** | **0.435** | **0.792** | **0.473** |
| | Promotion | 1.50% | 2.40% | 2.50% | 5.90% | 8.10% | 16.9% | 10.3% | 20.0% | 6.50% | 12.1% |
| GRU_96 | Spike-GRU | 0.243 | 0.924 | 0.240 | 0.919 | 0.213 | 0.932 | 0.191 | 0.944 | 0.171 | 0.970 |
| | **TS-GRU** | **0.342** | **0.857** | **0.341** | **0.860** | **0.338** | **0.863** | **0.319** | **0.869** | **0.294** | **0.878** |
| | Promotion | 40.7% | 7.80% | 42.0% | 6.86% | 58.7% | 7.99% | 67.0% | 8.63% | 71.9% | 10.4% |
| TCN_6 | Spike-TCN | 0.774 | 0.509 | 0.765 | 0.521 | 0.757 | 0.549 | 0.742 | 0.570 | 0.731 | 0.596 |
| | **TS-TCN** | **0.792** | **0.469** | **0.781** | **0.493** | **0.773** | **0.512** | **0.756** | **0.538** | **0.744** | **0.562** |
| | Promotion | 2.33% | 8.53% | 2.12% | 5.84% | 2.11% | 7.23% | 1.89% | 5.94% | 1.78% | 6.05% |
| TCN_96 | Spike-TCN | 0.315 | 0.884 | 0.307 | 0.914 | 0.248 | 0.936 | 0.214 | 0.973 | 0.202 | 0.996 |
| | **TS-TCN** | **0.323** | **0.870** | **0.319** | **0.883** | **0.276** | **0.914** | **0.256** | **0.925** | **0.228** | **0.970** |
| | Promotion | 2.54% | 1.61% | 3.91% | 3.52% | 11.3% | 2.41% | 19.6% | 5.19% | 12.9% | 2.68% |

## A.9 STANDARD DEVIATION ANALYSIS

Table 10 shows the standard deviation of $R^2$ and RSE metrics over three runs with different random seeds for TS-GRU and TS-former on the Metr-la and Electricity datasets, across prediction lengths of 6, 24, 48, and 96. Both models exhibit low standard deviations, demonstrating their stability and robustness. These results confirm the reliability of TS-GRU and TS-former under varying random seeds.

## A.10 AVERAGE POWER SPECTRUM ANALYSIS

To gain a deeper understanding of how TS-LIF processes temporal features, we analyze the average power spectrum of dendritic and somatic voltages after the first encoder layer of TS-former. Figure 7 illustrates the power distribution of voltage signals from dendrites and soma across different frequency ranges.

Table 10: The standard deviation of 3 runs with different random seeds with our TS-former and TS-GRU on Metr-la and Electricity datasets.

| Datasets | Metr-la | | | | | | | | Electricity | | | | | | | |
|---|---|---|---|---|---|---|---|---|---|---|---|---|---|---|---|---|
| Lengths | 6 | | 24 | | 48 | | 96 | | 6 | | 24 | | 48 | | 96 | |
| Metrics | $R^2\uparrow$ | RSE↓ | $R^2\uparrow$ | RSE↓ | $R^2\uparrow$ | RSE↓ | $R^2\uparrow$ | RSE↓ | $R^2\uparrow$ | RSE↓ | $R^2\uparrow$ | RSE↓ | $R^2\uparrow$ | RSE↓ | $R^2\uparrow$ | RSE↓ |
| TS-GRU (ours) | .002 | .005 | .004 | .006 | .002 | .011 | .004 | .009 | .001 | .002 | .001 | .005 | .002 | .004 | .001 | .008 |
| TS-former (ours) | .001 | .008 | .002 | .006 | .005 | .013 | .002 | .010 | .001 | .003 | .001 | .003 | .001 | .005 | .001 | .007 |

The analysis reveals distinct frequency response characteristics between dendritic and somatic compartments. These different roles allow TS-LIF to encode diverse temporal features effectively, contributing to its superior performance on time series forecasting tasks.

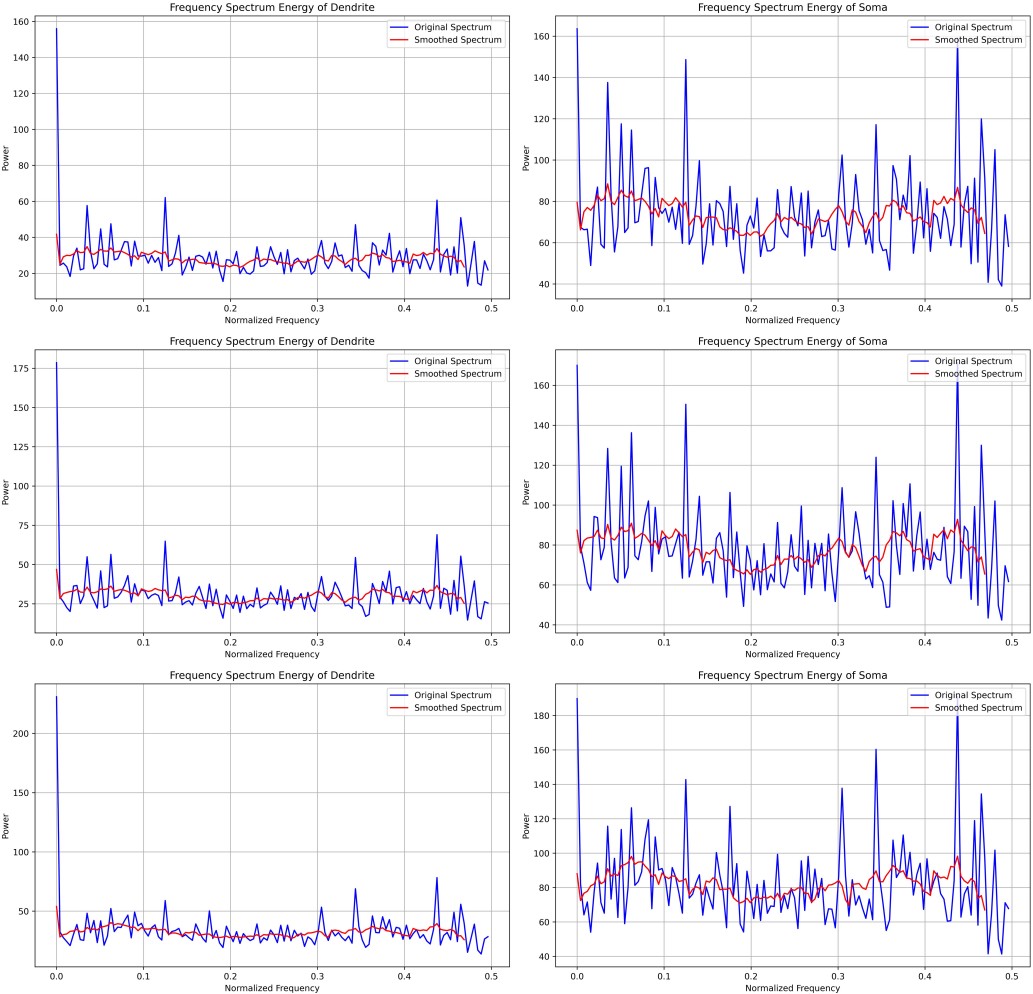

Figure 7: Average power spectrum analysis of dendritic and somatic voltages. The figure illustrates the power distribution of voltage signals from dendrites and soma across different frequency ranges, providing insights into neural signal processing mechanisms.

## A.11 Evaluating Long-Term Dependency Capture with Delayed Spiking XOR Problem

To further evaluate the ability of spiking neuron models to capture long-term dependencies, we conducted experiments using the delayed spiking XOR problem. This task tests the model's capacity to retain and process information over extended periods. The task involves three stages: an initial

spike input, a delay period with noisy spikes, and a second spike input. The model computes an XOR operation between the first and final inputs based on their firing rates.

Our experimental setup follows the parameters described in (Zheng et al., 2024). Specifically, we employed a two-layer MLP with only 20 hidden neurons, where the input feature size was set to 20. The results, measuring prediction accuracy under different delay timesteps and activation functions (ReLU, LIF, and TS-LIF), are summarized in Table 11.

Table 11: Prediction accuracy of the delayed spiking XOR problem under different delay timesteps and activation functions.

| Delay timesteps | ReLU | LIF | TS-LIF |
|---|---|---|---|
| 10 | 0.5 | 0.748 | 0.994 |
| 20 | 0.5 | 0.504 | 0.977 |
| 30 | 0.5 | 0.504 | 0.791 |
| 40 | 0.5 | 0.500 | 0.585 |
| 50 | 0.5 | 0.500 | 0.585 |

The results demonstrate that TS-LIF significantly outperforms both ReLU and standard LIF across all tested delay timesteps, particularly excelling at shorter and moderate delays. While the performance of LIF degrades as delay increases, TS-LIF retains higher accuracy, showcasing its enhanced capability for capturing long-term dependencies. This improvement can be attributed to the distinct processing mechanisms of dendritic and somatic compartments, which allow TS-LIF to maintain a more robust temporal memory compared to traditional spiking and non-spiking activation functions.

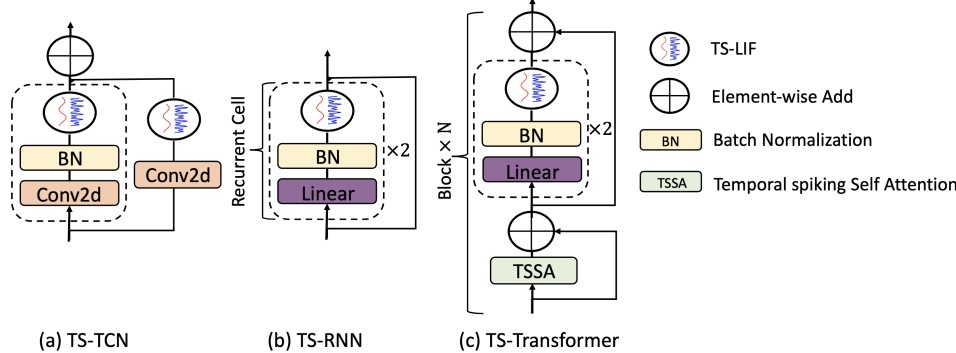

Figure 8: An illustration of our proposed TS-LIF based spiking structure in three models (TCN, RNN and Transformer).

## A.12  TEMPORAL ALIGNMENT AND SPIKE ENCODER

To utilize the intrinsic nature of SNN to its best, it's crucial to align the temporal dimension between time-series data and SNNs. Our central concept is to incorporate relevant finer information of the spikes within the time-series data at each time step. Specifically, we divide a time step $\Delta T$ of the time series into $T_s$ segments and each of them allows a firing event for neurons whose membrane potentials surpass the threshold, i.e., $\Delta T = T_s \Delta t$.

This equation bridges between a time-series time step $\Delta T$ and an SNN time step $\Delta t$. As a result, the independent variable $t$ in time-series ($\mathcal{X}(t)$) and in SNN ($U(t), I(t), H(t), S(t)$) are now sharing the same meaning. To this end, the spiking encoder, responsible for generating the first spike trains based on the floating-point inputs, needs to calculate $T_s \times T \times C$ possible spike events. The most straightforward non-parametric approach is to consider each data point in the input time series as the current value and replicate it $T_s$ times. However, this approach can disrupt the continuous nature of the underlying $\mathcal{X}(t)$ hypothesis. Therefore, we seek to use parametric spike encoding techniques.

Given the historical observed time-series $\mathbf{X} \in \mathbb{R}^{T \times C}$, we feed it into a convolutional layer followed by batch normalization and generate the spikes as:

$$\mathbf{S} = \mathcal{SNN}\left(\text{BN}\left(\text{Conv}\left(\mathbf{X}\right)\right)\right). \tag{23}$$

Where $\mathcal{SNN}$ is the TS-LIF neuron, by passing through the convolutional layer, the dimension of the spike train $\mathbf{S}$ is expanded to $T_s \times T \times C$. Spikes at every SNN time step are generated by pairing the data with different convolutional kernels.

The convolutional spike encoder capture internal temporal information of the input data, i.e., temporal changes and shapes, respectively, contributing to the representation of the dynamic nature of the information over time and catering to the following spiking layers for event-driven modeling. Also the pipeline of spiking based TCN, RNN-based models and Transformer strcuture are show in the

### A.13 LIMITATIONS AND FUTURE WORK

**Limitations**. In multivariate time series forecasting, modeling the correlations between variables is crucial for improving prediction accuracy. Current SNN-based models for time series forecasting primarily focus on temporal modeling and lack explicit mechanisms for capturing inter-variable correlations. For instance, explicitly computing cross-variable correlations, as shown in works like Ilbert et al. (2024a) and Zhang & Yan (2023), can effectively model multivariate relationships. We intend to explore how SNN filtering mechanisms can efficiently model cross-variable relationships to further enhance predictive performance.

**Future Work**. Future research directions include: (1) designing an efficient and effective SNN mechanism for capturing cross-variable correlations in multivariate time series, and (2) developing more generalized SNN structures for comprehensive time series analysis tasks, such as anomaly detection, time series generation, and classification.

