# OpenReview forum: "TS-LIF: A Temporal Segment Spiking Neuron Network for Time Series Forecasting"
_ICLR.cc/2025/Conference — ICLR 2025 Poster_

### Official Review · Reviewer_toyw · 2024-10-20

**Soundness:** 2
**Presentation:** 3
**Contribution:** 2
**Rating:** 6
**Confidence:** 4

**Summary:**

The authors propose an extension of the dual-compartment Leaky Integrate-and-Fire model with the aim of improving time-series forecasting, by capturing both low- and high-frequency input in their model. The authors validate the prediction accuracy of their model compared to other models on different datasets and demonstrate that their model performs slightly better.

**Strengths:**

The proposed model is interesting, and the paper is generally well-written. In addition, the authors include several lines of analysis and benchmark their model on four different timeseries datasets - the preliminaries section is also well written.

**Weaknesses:**

I have several concerns regarding the work.

1. **Questionable biological realism.** The proposed idea of modeling the voltage of the dendrites and soma separately is interesting (although not new [1]). However, adding the dendritic and somatic spike outputs using a weighted sum (Eq. 6) is biologically questionable, as the outputs of the units now become graded and are no longer spiking. How is this modeling step reconciled in biology?  If the model is intended to capture the biology, then I would expect to see more comparisons to actual electrophysiological recordings of neurons.

2. **Is the model more efficient than other neural networks?** You state that TCNs, Autofromers, and Transformers require more memory and computational resources (stated on line 49 and 145-148). Does your model overcome this issue? I would assume that your model would also incur significant resource demands during training. Furthermore, it is not clear how your model could be ported to a neuromorphic computer for inference, as the output is not strictly spiking.

3. **Missing controls.** The author's motivation for the work is that spiking neurons struggle to capture long-term dependencies in temporal input. However, they do not mention (or compare to) works using trainable membrane time constants [2] or trainable adaptive firing rates [3], which have been shown to significantly improve SNN performance on different time-series datasets. Simply making membrane time constant trainable can even make single-spike SNNs outperform standard multi-spike LIF-SNNs on complex temporal datasets [4]. I would suggest the authors compare their model to models in which the membrane time constant and adaptive firing rates are learned.

4. **Confusion regarding the analysis.** In the stability analysis (Sec. 4.2) the authors use the equations of the dual comportment model (Eq. 4) rather than their model (Eq. 5). Why was this done? Furthermore, the analysis was only conducted on the homogenous part of the equations, please explain why this assumption can be made. I am also unsure why this analysis was done, as – unlike standard artificial neural networks – the output of spiking neurons is bounded, so why would we need to analyze these conditions? Lastly, where did you utilize this analysis in your work (e.g. for setting hyperparameters or initial weights)?

[1] Gerstner, W., Kistler, W.M., Naud, R. and Paninski, L., 2014. Neuronal dynamics: From single neurons to networks and models of cognition. Cambridge University Press.

[2] Perez-Nieves, N., Leung, V.C., Dragotti, P.L. and Goodman, D.F., 2021. Neural heterogeneity promotes robust learning. Nature communications, 12(1), p.5791.

[3] Yin, B., Corradi, F. and Bohté, S.M., 2021. Accurate and efficient time-domain classification with adaptive spiking recurrent neural networks. Nature Machine Intelligence, 3(10), pp.905-913.

[4] Taylor, L., King, A. and Harper, N., 2022. Robust and accelerated single-spike spiking neural network training with applicability to challenging temporal tasks. arXiv preprint arXiv:2205.15286.

**Questions:**

### Questions:
- What is the reason for introducing equation 3 on page 4? This was not used anywhere else.
- I am not familiar with the frequency response analysis (Sec. 4.3). Perhaps you could provide more background on what this is measuring, why this was done, and where it is used. For example, how do we interpret the values of H_d(z) and H_s(d)? Shouldn’t the low frequencies and high-frequencies have units Hz (line 341-343)? Also, why is w=pi equal to high-frequency?
- What are the top row numbers in Table 1? Are these L? If not, what are they and what was L set to?
- Why does your model outperform the Transformer architecture (e.g. iTransformer)? SNNs do not usually outperform non-spiking neural networks. Please try to discuss this.
- Can you also discuss why your model might perform better on certain datasets? i.e. how certain frequency information is discarded by the other models.
- Did you use surrogate gradients for training?
- In Figure 2, what do training timesteps refer to? Is this the number of training epochs or gradient updates?
- In the robustness evaluation you only use the Electricity dataset. However, this dataset seems to be “easily solved” using other models under different ratios of missing values, making it hard to infer if your model is better – perhaps you could run your analysis on a harder dataset (e.g. Metr-la)?
- What are the main limitations of your work – I would suggest adding a discussion to your paper.
- Is the code publicly available?
### Additional feedback to improve your paper:
- I would suggest moving the Temporal Analysis (Sec. 5.3.1) before the benchmarking section, as this allows the reader to get a better feel for the model early on.
- If possible, I would include multiple training runs with different random seeds and include some statistical analysis to better compare the results in Table 1, as many of the results are very similar, making it hard to know if this is due to weight initialization or not. Also, what are the shaded regions in Fig. 2 (eg. SD or SEM)?
- Typo in Fig. 2 plot 2 y-label.
- Missing reference for iTransformer (line 378).

---

> ### Author Response · Authors · 2024-11-23
> **Response to Reviewer toyw (Part 1)**
>
> Thank you for taking the time to review our paper and providing valuable feedback. We would like to answer your questions below.
>
> **W1: Questionable biological realism**
>
> **WA1:** Thank you for acknowledging the value of modeling dendritic and somatic voltages separately. We address your concern as follows:
> 1. The weighted summation in our model can be interpreted as representing the influence weights of two distinct types of spikes on the next layer of neurons, akin to synaptic connection strength. This weight, $\kappa$, can be seamlessly integrated into the weights of the next layer (e.g., MLP) without altering the binary spiking nature (0 or 1) of the outputs.
>
> 2. From another perspective, the dendrites and soma in our TS-LIF model can be viewed as interacting units that process information at different frequencies while influencing one another. This is analogous to sensory systems, such as auditory processing, where inner hair cells are tuned to different characteristic frequencies, or the visual system, where cone cells are tuned to different wavelengths. We believe that this perspective adds biological credibility to our approach.
>
> A more detailed biological explanation is provided in the revised manuscript under the **Temporal Segment LIF Neuron** section. The modifications have been highlighted to clarify this connection and further emphasize the conceptual similarities.
>
> **W2: Is the model more efficient than other neural networks?**
>
> **WA2:** We apologize for any misunderstanding caused. Our original intention was to emphasize that TCN and transformer require more memory compared to GRU, rather than comparing them to SNN. The efficiency of SNNs primarily lies in their ability to significantly reduce energy consumption. Compared to traditional ANNs, SNNs reduce energy consumption by approximately 60%, as shown in the table below:
>
> **Table. The theoretical energy consumption and performance comparison during the inference stage of the Electricity dataset**
> |**Model**|**Param(M)**|**OPs(G)**|**Energy(mJ)**|**EnergyReduction**|**R²**|
> |-|-|-|-|-|-|
> |**TCN**|0.460|0.14|0.64|-|**.973**|
> |**TS-TCN(Ours)**|0.465|0.19|0.25|60.93%↓|.971|
> |**GRU**|1.288|1.32|6.07|-|.972|
> |**TS-GRU(Ours)**|1.291|1.67|1.58|73.80%↓|**.981**|
> |**iTransformer**|1.634|2.05|9.47|-|.977|
> |**TS-former(Ours)**|1.640|3.59|3.22|65.99%↓|**.985**|
>
> **W3: Missing controls.**
>
> **WA3:** We believe there may have been a misunderstanding regarding the parameters in our TS-LIF model. All parameters of the TS-LIF model are learnable. To address this potential confusion, we have clarified this point in the revised manuscript.
>
> Additionally, to further validate the TS-LIF model's ability to capture **long-term dependencies**, we conducted additional experiments using the **Delayed Spiking XOR Problem**. This task involves three stages: an initial spike input, a delay period with noisy spikes, and a second spike input. The model computes an XOR operation between the first and final inputs based on firing rates.
>
> Our experimental parameters and setup align with those described in [1]. Specifically, we used a **two-layer MLP** with only 20 hidden neurons, where the input feature size was set to 20. The results, which measure prediction accuracy under different delay timesteps and activation functions, are summarized as follows:
>
> **Table. The prediction accuracy of the delayed spiking XOR problem under different delay timesteps and activation functions.**
> |Delay timesteps|ReLU|LIF|**TS-LIF**|
> |-|-|-|-|
> |10|0.500|0.748|**0.994**|
> |20|0.500|0.504|**0.977**|
> |30|0.500|0.504|**0.791**|
> |40|0.500|0.500|**0.585**|
> |50|0.500|0.500|**0.585**|
> - "Delay timesteps" refers to the delay period with noisy spikes.
> - "ReLU" denotes the activation function
> - "LIF" denotes the vanilla LIF neuron.
> - "TS-LIF" denotes our proposed TS-LIF neuron.
>
> The results clearly demonstrate the superior performance of the TS-LIF model in capturing long-term dependencies compared to both ReLU and  vanilla LIF activation functions.
>
> Additionally, to verify the **effectiveness** of our TS-LIF, we extended the comparison to include TC-LIF [2], LM-H [3], and CLIF [4] in the GRU Backbone with three prediction lengths (6, 24, 96) of two datasets (Metr-la and Electricity) , where TS-LIF consistently outperformed these models. The detailed results are summarized below:
>
> **Table. Comparison of TS-LIF with Other LIF Neurons**
> |Datasets|Metr-la||||||Electricity||||||
> |-|-|-|-|-|-|-|-|-|-|-|-|-|
> |Lengths|6||24||96||6||24||96|||
> |Metrics|R²↑|RSE↓|R²↑|RSE↓|R²↑|RSE↓|R²↑|RSE↓|R²↑|RSE↓|R²↑|RSE↓|
> |TC-LIF|0.828|0.453|0.594|0.673|0.259|0.956|0.978|0.263|0.976|0.257|0.941|0.503|
> |LM-H|0.812|0.464|0.570|0.719|0.246|0.973|0.971|0.269|0.969|0.280|0.936|0.512|
> |CLIF|0.837|0.429|0.606|0.667|0.271|0.930|0.973|0.259|0.972|0.276|0.954|0.376|
> |**TS-LIF**|**0.848**|**0.412**|**0.618**|**0.651**|**0.329**|**0.853**|**0.991**|**0.216**|**0.981**|**0.240**|**0.976**|**0.271**|

---

> > ### Author Response · Authors · 2024-11-23
> > **Response to Reviewer toyw (Part 2)**
> >
> > **W4: Confusion regarding the analysis.**
> >
> > **WA4**: Thank you for raising these concerns about our stability analysis. We address your points as follows:
> > 1. **Why was Eq.(4) used instead of Eq.(5) ?**
> > Both Eq.(4) and Eq.(5) describe the dual compartment model, but their homogeneous parts are equivalent and can be represented by Eq.(7). Therefore, our analysis focuses on Eq.(7) to study the intrinsic dynamics of the system.
> >
> > 2. **Why was the analysis conducted only on the homogeneous part?**
> > Focusing on the homogeneous part is a standard approach in stability analysis, as a system's stability is fundamentally determined by its intrinsic dynamics (e.g., eigenvalues and time constants) rather than external inputs. Inputs are task-specific and transient.
> >
> > 3. **Why conduct stability analysis for spiking neurons with bounded outputs?**
> > While spiking neurons have bounded outputs (binary spikes), their internal states, such as membrane potentials, can still diverge. Stability analysis ensures that these internal states remain bounded, preserving the model's ability to learn and perform reliably. Additionally, this analysis provides a foundation for understanding the interplay between dendritic and somatic dynamics, further supporting the model’s design.
> >
> > 4. **How was this analysis utilized in the work?**
> > The stability analysis directly influenced the selection of model hyperparameters, such as the decay constants in the dendritic and somatic compartments, ensuring the model operates in a stable regime. It also guided the initialization of key parameters, such as time constants and weights, to facilitate stable training and convergence.
> >
> > **Q1: The reason for introducing Equation 3**
> >
> > **A1**: Equation 3 is introduced to demonstrate that traditional LIF neurons can be viewed as performing a moving average or low-pass filtering operation in the temporal domain.
> >
> > **Q2: The frequency response analysis**
> >
> > **A2**: The magnitudes of $H_d(z)$ and $H_s(z)$ represent the system's gain at specific normalized frequencies, while the phase indicates the delay or shift introduced by the system. In the discrete-time frequency domain, normalized angular frequency ($\omega$) is expressed in radians per sample, ranging from $0$ to $\pi$, with $\omega = \pi$ corresponding to the Nyquist frequency—the highest frequency representable in a discrete system. To convert **normalized angular frequency $\omega$** to **physical frequency** in Hertz (Hz), the following formula is used: $f = \frac{\omega}{2\pi} f_s$ where $f_s$ is the sampling frequency in Hz. This conversion allows normalized frequencies to be interpreted in real-world units.
> >
> > **Q3: What are the top row numbers in Table 1? Are these L? If not, what are they and what was L set to?**
> >
> > **A3**: The top row numbers in Table 1 represent L, the prediction horizons, which are the lengths of the prediction sequences.
> >
> > **Q4: Outperform the Transformer architecture?**
> >
> > **A4**: The activation functions of LIF and TS-LIF neurons, which have stronger temporal modeling capabilities compared to ReLU.
> >
> > **Q5: Can you also discuss why your model might perform better on certain datasets?**
> >
> > **A5**: The performance is influenced by both the characteristics of the dataset and the model's design. For instance, **TS-TCN** performs particularly well on the Pems-bay dataset, while **TS-former** excels on the Solar and Electricity datasets. The alignment between each model's architecture and the specific dynamics of the datasets plays a crucial role in determining performance.
> >
> > **Q6: Use surrogate gradients?**
> >
> > **A6**: Yes, this is the standard practice for training SNN models and we follow it.
> >
> > **Q7: In Figure 2, what do training timesteps refer to?**
> >
> > **A7**: Training timesteps refer to the length of the historical sequence used for training.

---

> > > ### Author Response · Authors · 2024-11-23
> > > **Response to Reviewer toyw (Part 3)**
> > >
> > > **Q8: robustness evaluation on Metr-la**
> > >
> > > **A8**: We conducted a robustness evaluation on Metr-la, and the results demonstrate that our model remains the best. Detailed results are provided in the newly submitted **Appendix Table 9**.
> > >
> > > **Table 9: Experimental performance of the **TS-LIF (ours)** models (TS-former, TS-GRU, TS-TCN) compared to the vanilla LIF on the Metr-la dataset, evaluated under different ratios of missing values in the historical inputs**.
> > >
> > > | **Missing Ratio** || **10%** || **20%**|| **40%**|| **60%**|| **80%**||
> > > |-|-|-|-|-|-|-|-|-|-|-|-|
> > > ||| **R$^2$$\uparrow$**| **RSE$\downarrow$**| **R$^2$$\uparrow$**| **RSE$\downarrow$**| **R$^2$$\uparrow$**| **RSE$\downarrow$**| **R$^2$$\uparrow$**| **RSE$\downarrow$**| **R$^2$$\uparrow$**| **RSE$\downarrow$**|
> > > | **Transformer\_6** | iSpikformer| 0.815| 0.479| 0.813| 0.486| 0.809 | 0.488| 0.804| 0.492| 0.802 | 0.496|
> > > | | **TS-former** | **0.843** | **0.419** | **0.842**| **0.421** | **0.838** | **0.430**| **0.835** | **0.436**| **0.831**| **0.440**|
> > > || **Promotion**| 3.43%| 14.3%| 3.56%| 15.4%| 3.58%| 13.4%| 3.86%| 12.8%| 3.61%| 12.7%|
> > > | **Transformer\_96**| iSpikformer| 0.270| 0.915| 0.254| 0.926| 0.230 | 0.935| 0.204| 0.948| 0.194| 0.959|
> > > || **TS-former**| **0.277**| **0.907**| **0.263**| **0.911**| **0.238**| **0.927**| **0.212**| **0.936**| **0.205** | **0.945**|
> > > || **Promotion**| 2.59%| 0.88%| 3.54%| 1.65%| 3.47%| 0.86%| 3.92%| 1.28% | 5.67%| 1.48%|
> > > | **GRU\_6**| Spike-GRU| 0.830| 0.429| 0.819| 0.440| 0.771| 0.497| 0.746 | 0.522| 0.743| 0.530 |
> > > || **TS-GRU**| **0.843**| **0.417** | **0.839**| **0.414**| **0.834**| **0.425**| **0.823**| **0.435**| **0.792**| **0.473** |
> > > || **Promotion**| 1.50%| 2.40%| **2.50%**| **5.90%**| **8.10%** |**16.9%**| **10.3%**| **20.0%**| **6.50%** |**12.1%**|
> > > | **GRU\_96**| Spike-GRU | 0.243  | 0.924| 0.240| 0.919| 0.213 | 0.932 | 0.191| 0.944| 0.171| 0.970|
> > > || **TS-GRU**| **0.342**| **0.857**| **0.341**| **0.860**| **0.338**| **0.863**| **0.319**| **0.869**| **0.294**| **0.878**|
> > > || **Promotion** | **40.7%**|**7.80%**|**42.0%**|**6.86%**|**58.7%**| **7.99%**|**67.0%**|**8.63%**|**71.9%**|**10.4%**|
> > > | **TCN\_6** | Spike-TCN| 0.774 | 0.509| 0.765| 0.521| 0.757| 0.549| 0.742| 0.570 | 0.731| 0.596|
> > > || **TS-TCN** | **0.792**| **0.469**| **0.781** | **0.493**| **0.773**| **0.512**| **0.756**| **0.538**| **0.744**| **0.562**|
> > > || **Promotion**| 2.33%| 8.53%| 2.12%| 5.84%| 2.11%| 7.23%| 1.89%| 5.94%| 1.78%| 6.05%|
> > >
> > > - Model\_* indicates a backbone model with a prediction length of *, such as Transformer\_6 represents the Transformer architecture with a prediction length of 6.
> > >
> > > Due to time constraints, the TCN_96 experiment is still ongoing. We will continue to provide updates in the future. Thank you for your understanding.
> > >
> > > **Q9: Limitations of your work**
> > >
> > > **A9**: The limitations of our work are discussed in the Appendix in the submission.
> > >
> > > **Q10: Is the code publicly available?**
> > >
> > > **A10**: Yes, the code will be made publicly available.
> > >
> > > **S1: Moving the Temporal Analysis (Sec. 5.3.1) before the benchmarking section**
> > >
> > > **SA1**: In the revised version of our submission, we have adjusted the order accordingly.
> > >
> > > **S2: Multiple training runs with different random seeds**
> > >
> > > **SA2**: We further report the standard deviation across three runs with different random seeds for our TS-former and TS-GRU models on the Metr-la and Electricity datasets.
> > >
> > > **Table. The standard deviation of 3 runs with different random seeds with our TS-former and TS-GRU on Metr-la and Electricity datasets**.
> > > |Datasets|Metr-la||||||||Electricity||||||||
> > > |-|-|-|-|-|-|-|-|-|-|-|-|-|-|-|-|-|
> > > |Lengths|6||24||48||96||6||24||48||96|
> > > ||R²↑|RSE↓|R²↑|RSE↓|R²↑|RSE↓|R²↑|RSE↓|R²↑|RSE↓|R²↑|RSE↓|R²↑|RSE↓|R²↑|RSE↓|
> > > |**TS-GRU**|0.002|0.005|0.004|0.006|0.002|0.011|0.004|0.009|0.001|0.002|0.001|0.005|0.002|0.004|0.001|0.008|
> > > |**TS-former**|0.001|0.008|0.002|0.006|0.005|0.013|0.002|0.010|0.001|0.003|0.001|0.003|0.001|0.005|0.001|0.007|
> > >
> > > Due to time constraints, the remaining experimental results will be continuously updated. Thank you for your understanding.
> > >
> > > **S3: Missing reference for iTransformer (line 378).**
> > >
> > > **SA3**: We have updated the revised version to include the appropriate reference for iTransformer.
> > >
> > > We sincerely hope our clarifications above have addressed your concerns and can improve your opinion of our work.
> > >
> > > [1] Zheng H, Zheng Z, Hu R, et al. Temporal dendritic heterogeneity incorporated with spiking neural networks for learning multi-timescale dynamics[J]. Nature Communications, 2024, 15(1): 277.
> > >
> > > [2] TC-LIF: A Two-Compartment Spiking Neuron Model for Long-Term Sequential Modelling, AAAI 2024
> > >
> > > [3] A Progressive Training Framework for Spiking Neural Networks with Learnable Multi-hierarchical Model, ICLR 2024
> > >
> > > [4] CLIF: Complementary Leaky Integrate-and-Fire Neuron for Spiking Neural Networks, ICML 2024

---

> > > > ### Author Response · Authors · 2024-11-25
> > > > **Response to Reviewer toyw (Part 4)**
> > > >
> > > > **Q8: robustness evaluation on Metr-la.** (Updated Experimental Results)
> > > >
> > > > **A8**: The following Table 10 presents the experimental results for TCN_96, which were previously incomplete in Table 9.
> > > > ### Table 10: Experimental performance of the **TS-TCN (ours)** models compared to the vanilla LIF based TCN (Spike-TCN) on the Metr-la dataset, evaluated under different ratios of missing values in the historical inputs.
> > > > *Model\_* indicates a backbone model with a prediction length of *, such as TCN_96 represents the Transformer architecture with a prediction length of 96.*
> > > > | **Missing Ratio** || **10%** || **20%**|| **40%**|| **60%**|| **80%**||
> > > > |-|-|-|-|-|-|-|-|-|-|-|-|
> > > > ||| **R$^2$$\uparrow$**| **RSE$\downarrow$**| **R$^2$$\uparrow$**| **RSE$\downarrow$**| **R$^2$$\uparrow$**| **RSE$\downarrow$**| **R$^2$$\uparrow$**| **RSE$\downarrow$**| **R$^2$$\uparrow$**| **RSE$\downarrow$**|
> > > > | **TCN\_96** | Spike-TCN| 0.315 |0.884|0.307|0.914|0.248|0.936|0.214|0.973|0.202|0.996|
> > > > || **TS-TCN** | **0.323**| **0.870**| **0.319** | **0.883**| **0.276**| **0.914**| **0.256**| **0.925**| **0.228**| **0.970**|
> > > > || **Promotion**| 2.54%| 1.61%| 3.91%| 3.52%| 11.3%| 2.41%| 19.6%|5.19%| 12.9%| 2.68%|
> > > >
> > > > If you have any further questions, we are looking forward to discussing with you.

---

> > > > > ### Author Response · Authors · 2024-11-26
> > > > > **Kindly Request for Reviewer's Feedback**
> > > > >
> > > > > Dear reviewer toyw,
> > > > >
> > > > > Thank you for your time and effort in providing such valuable feedback on our work. As the discussion period is nearing its end, we hope you’ve had the chance to go through our rebuttal. We believe it has helped clarify our work further, and if our responses have addressed your concerns, we would greatly appreciate it if you could kindly update your feedback to reflect this. We’re more than happy to continue the discussion if there are any remaining questions or points to address.
> > > > >
> > > > > Sincerely
> > > > >
> > > > > Authors

---

> > > > > > ### Comment · Reviewer_toyw · 2024-11-26
> > > > > >
> > > > > > Thank you for taking the time to address all of my comments and questions! The authors have put a significant amount of time into addressing my concerns and I have increased my rating.

---

> > > > > > > ### Author Response · Authors · 2024-11-26
> > > > > > > **Response to Reviewer toyw**
> > > > > > >
> > > > > > > Dear reviewer toyw
> > > > > > >
> > > > > > > Thank you for your kind feedback and for recognizing our efforts in addressing your comments and questions. We value your thoughtful comments and are glad that our response has resolved your concerns. Your comments play an important role in improving the quality of our work.
> > > > > > >
> > > > > > > Sincerely,
> > > > > > >
> > > > > > > Authors

---

### Official Review · Reviewer_a5wF · 2024-10-29

**Soundness:** 2
**Presentation:** 2
**Contribution:** 2
**Rating:** 6
**Confidence:** 3

**Summary:**

To enhance the ability of SNNs to model long-term dependencies and process multi-scale temporal information, this paper proposes the Temporal Segment LIF (TS-LIF) model, which builds upon the existing Two-Compartment LIF (TC-LIF) model. The two compartments in TS-LIF are specifically designed to process low- and high-frequency signals, respectively. Theoretical analysis demonstrates the stability of the TS-LIF model, and experiments on time series forecasting tasks show that TS-LIF outperforms the standard LIF model.

**Strengths:**

- Effectively modeling long-range dependencies remains an important yet unresolved challenge in SNNs.

- Using two compartments to separately capture low- and high-frequency information is novel.

**Weaknesses:**

- The primary concern lies in the experimental design. This paper evaluates the TS-LIF model by integrating it with TCN, GRU, and Transformer architectures on time series forecasting tasks. However, it is questionable whether these setups and datasets effectively assess the ability of spiking neuron models to capture long-term dependencies. Given that GRU, TCN, and Transformers already have strong long-sequence modeling capabilities, the performance of SNNs may not be adequately represented.

- The paper lacks comparisons with existing spiking neuron models such as TC-LIF [1], LM-H [2], CLIF [3], and BHRF [4].

- It is biologically questionable whether the input current can be applied to both the dendrite and soma while producing weighted spikes.

[1] TC-LIF: A Two-Compartment Spiking Neuron Model for Long-Term Sequential Modelling, AAAI 2024

[2] A Progressive Training Framework for Spiking Neural Networks with Learnable Multi-hierarchical Model, ICLR 2024

[3] CLIF: Complementary Leaky Integrate-and-Fire Neuron for Spiking Neural Networks, ICML 2024

[4] Balanced Resonate-and-Fire Neurons, ICML 2024

**Questions:**

- How many time steps are used in TS-LIF on the time series forecasting tasks?

- What are the specific values of the hyperparameters used in TS-LIF on the time series experiments? Are there established principles for determining these hyperparameters, and what effects do they have on model performance?

- What is the value of the decay factor in the LIF baselines presented in Table 1? Have you examined the influence of the decay factor on performance in these time series tasks?

---

> ### Author Response · Authors · 2024-11-23
> **Response to Reviewer a5wF (Part 1)**
>
> Thank you for taking the time to review our paper and providing valuable feedback. We would like to answer your questions below.
>
> **W1: The ability of spiking neuron models to capture long-term dependencies**
>
> **WA1**: In the submission, we compared three different types of sequence models. Among them, GRU lacks the ability to model long-term dependencies, whereas TS-LIF consistently demonstrated superior performance. This indicates that TS-LIF effectively enhances the model's capability for long-term modeling.
>
> In the rebuttal, to further evaluate the ability of spiking neuron models to capture long-term dependencies, we conducted experiments using the **Delayed Spiking XOR Problem**. This task involves three stages: an initial spike input, a delay period with noisy spikes, and a second spike input. The model computes an XOR operation between the first and final inputs based on firing rates. Our experimental parameters and setup align with those described in [1]. Specifically, we use a **two-layer MLP** with only 20 hidden neurons, where the input feature size was set to 20. The results, which measure prediction accuracy under different delay timesteps and activation functions, are summarized as follows:
>
> **Table. The prediction accuracy of the delayed spiking XOR problem under different delay timesteps and activation functions.**
> |Delay timesteps|ReLU|LIF|TS-LIF|
> |-|-|-|-|
> |10|0.500|0.748|**0.994**|
> |20|0.500|0.504|**0.977**|
> |30|0.500|0.504|**0.791**|
> |40|0.500|0.500|**0.585**|
> |50|0.500|0.500|**0.585**|
> - "Delay timesteps" refers to the delay period with noisy spikes.
> - "ReLU" denotes the activation function
> - "LIF" denotes the vanilla LIF neuron.
> - "TS-LIF" denotes our proposed TS-LIF neuron.
>
> The results clearly demonstrate the superior performance of the TS-LIF model in capturing long-term dependencies compared to both ReLU and LIF activation functions.
>
> **W2: Comparisons with existing spiking neuron models such as TC-LIF, LM-H, CLIF, and BHRF.**
>
> **WA2**: In the **Ablation Study** section of our submission, we have compared the performance of TS-LIF and TC-LIF, demonstrating that TS-LIF delivers superior results. Additionally, we extended the comparison to include LM-H and CLIF in the GRU Backbone (noting that BHRF does not provide publicly available code) on two different types of datasets (Metr-la and Electricity), where TS-LIF consistently outperformed these models. The detailed results are summarized below:
> |Datasets|Metr-la||||||Electricity||||||
> |-|-|-|-|-|-|-|-|-|-|-|-|-|
> |Lengths|6||24||96||6||24||96|||
> |Metrics|R²↑|RSE↓|R²↑|RSE↓|R²↑|RSE↓|R²↑|RSE↓|R²↑|RSE↓|R²↑|RSE↓|
> |TC-LIF|0.828|0.453|0.594|0.673|0.259|0.956|0.978|0.263|0.976|0.257|0.941|0.503|
> |LM-H|0.812|0.464|0.570|0.719|0.246|0.973|0.971|0.269|0.969|0.280|0.936|0.512|
> |CLIF|0.837|0.429|0.606|0.667|0.271|0.930|0.973|0.259|0.972|0.276|0.954|0.376|
> |**TS-LIF**|**0.848**|**0.412**|**0.618**|**0.651**|**0.329**|**0.853**|**0.991**|**0.216**|**0.981**|**0.240**|**0.976**|**0.271**|
>
> **W3: It is biologically questionable whether the input current can be applied to both the dendrite and soma while producing weighted spikes.**
>
> **WA3:** We address this concern as follows:
> 1. In the **Hierarchical Temporal Memory (HTM)** neuron model, numerous synapses distributed along the dendrites function as pattern detectors that trigger NMDA spikes and depolarization at the soma [2]. This aligns closely with the TS-LIF model's approach, which leverages dendritic and somatic integration to process inputs across **diverse temporal patterns**. Furthermore, the TS-LIF model’s ability to generate both dendritic and somatic spikes is inspired by the biological capability of neurons to simultaneously process and predict signals, similar to the prediction mechanisms observed in HTM neurons.
> 2. The **weighted summation of dendritic and somatic spikes** in our model serves to reflect their differential influence on the subsequent layer of neurons, akin to the varying synaptic connection strengths observed in biological systems. This weighted influence can be integrated into the synaptic weights in the next layer (such as MLP).
> 3. Furthermore, from another perspective, dendrites and soma in our TS-LIF model can be viewed as interacting units, each processing different aspects of input information while influencing each other. This approach finds parallels in biological sensory systems, such as the auditory system, where inner hair cells are tuned to different characteristic frequencies, and the visual system, where cone cells are tuned to different wavelengths. These biological examples support the biological credibility of our TS-LIF model's structure and the **interaction between dendritic and somatic compartments**.
>
> A more detailed biological explanation is provided in the revised manuscript under the **Temporal Segment LIF Neuron** section. The modifications have been highlighted to clarify this connection and further emphasize the conceptual similarities.

---

> > ### Author Response · Authors · 2024-11-23
> > **Response to Reviewer a5wF (Part 2)**
> >
> > **Q1: How many time steps are used in TS-LIF for the time series forecasting tasks?**
> >
> > **A1**: For a fair comparison, our TS-LIF is consistent with [3] and we use time steps T=4.
> >
> > **Q2: What are the specific values of the hyperparameters used in TS-LIF for the time series experiments?**
> >
> > **A2**: All hyperparameters are learnable, including $\alpha$, $\beta$, $\gamma$, and $\kappa$ (where $\kappa$ is a vector corresponding to the feature's dimensions, while $\alpha$, $\beta$, and $\gamma$ are scalars). In the latest version, we provided a clearer description of this concept at **line 225**. Thank you for helping us improve the readability of our TS-LIF.
> >
> > **Q3: What is the value of the decay factor in the LIF baselines presented in Table 1?**
> >
> > **A3**: The results for the LIF baselines in Table 1 are directly sourced from [3]. Therefore, the decay factor was not set by us but remains consistent with [3].
> >
> > We sincerely hope our clarifications above have addressed your concerns and can improve your opinion of our work.
> >
> > [1] Zheng H, Zheng Z, Hu R, et al. Temporal dendritic heterogeneity incorporated with spiking neural networks for learning multi-timescale dynamics[J]. Nature Communications, 2024, 15(1): 277.
> >
> > [2] Hawkins J, Ahmad S. Why neurons have thousands of synapses, a theory of sequence memory in neocortex[J]. Frontiers in neural circuits, 2016, 10: 174222.
> >
> > [3] Lv C, Wang Y, Han D, et al. Efficient and Effective Time-Series Forecasting with Spiking Neural Networks[C]//Forty-first International Conference on Machine Learning.

---

> > > ### Comment · Reviewer_a5wF · 2024-11-25
> > >
> > > The reviewer appreciates the authors' detailed response and the additional comparative experiments, which effectively demonstrate the performance improvements of TS-LIF over other spiking neuron models on the time series task. However, given that TS-LIF is intended to enhance the modeling of long-term dependencies and multi-scale dynamics, the reviewer remains concerned that using only 4 time steps in most experiments is insufficient to demonstrate the model's claimed ability to capture long-term dependencies. Therefore, the reviewer remains inclined to reject and maintains the original score.

---

> > > > ### Author Response · Authors · 2024-11-25
> > > > **Response to Reviewer a5wF**
> > > >
> > > > Thanks again for your valuable feedback. There might be some misunderstanding here.
> > > >
> > > > To ensure a fairer comparison between TS-LIF and [3], we followed the experimental setup outlined in [3]. Given history inputs $X$ ={$x_1$, $x_2$, ..., $x_T$} $\in$ $R^{T \times D} $ with $T$ history lengths and $D$ variates. We follow the setting of 3.2 section **Temporal Alignment and Spike Encoder** in [3], which use different convolutional kernels to transform the history inputs $X$ $\in$ $R^{T \times D}$ into $\hat{X}$ $\in$ $R^{T_s \times T \times D}$, where the $T_s$ denotes the time steps. For example, the backbone is GRU, history length **$T$**=**96** and **$T\_s$=4**. At each time step $t$ within $T$, TS-LIF executes $T_s$ times. Therefore, the total actual steps experienced by TS-LIF neurons amount to **4 $\times$ 96**, rather than just 4 for the entire $T$ sequence. In this time series forecasting setup, capturing long-term dependencies is crucial. Figures 4 and 5 in our manuscript demonstrate TS-LIF's ability to capture multi-scale information and its performance advantages.
> > > >
> > > > Additionally, when comparing TS-LIF with TC-LIF, LM-H, and CLIF, it also shows superiority on both the Metr-la and Electricity datasets with 3 prediction lengths (6, 24, 96).
> > > >
> > > > We sincerely apologize for any misunderstanding caused by our explanation and appreciate your patience and feedback.
> > > >  If you have any further questions, we are looking forward to discussing with you.
> > > >
> > > > [3] Lv C, Wang Y, Han D, et al. Efficient and Effective Time-Series Forecasting with Spiking Neural Networks[C]//Forty-first International Conference on Machine Learning.

---

> > > > > ### Comment · Reviewer_a5wF · 2024-11-25
> > > > >
> > > > > The sequence processing mechanism remains unclear to the reviewer. Is the membrane potential of TS-LIF continuously integrated over $4\times96$ time steps, or is it reset to zero every 4 time steps? Could the authors provide more details about the temporal alignment and the spike encoder used in the experiments? Why are additional $T_s$ time steps introduced instead of directly using $T$?

---

> > > > > > ### Author Response · Authors · 2024-11-26
> > > > > > **Response to Reviewer a5wF**
> > > > > >
> > > > > > Thanks again for your constructive feedback, and sorry for delay response.
> > > > > >
> > > > > > Q1: Is the membrane potential of TS-LIF continuously integrated over $4 \times 96 $ time steps, or is it reset to zero every 4 time steps
> > > > > >
> > > > > > A1: The membrane potential of TS-LIF will continuously integrated over $4 \times 96 $ time steps and then reset.
> > > > > >
> > > > > > Q2: Could the authors provide more details about the temporal alignment and the spike encoder used in the experiments?
> > > > > >
> > > > > > A2: The **temporal alignment and the spike encoder** is source from [3], and detailed descriptions are shown in the following part.
> > > > > >
> > > > > > **Temporal alignment**: To utilize the intrinsic nature of SNN to its best, it's crucial to align the temporal dimension between time-series data and SNNs. Our central concept is to incorporate relevant finer information of the spikes within the time-series data at each time step. Specifically, we divide a time step $\Delta T$ of the time series into $T_s$ segments and each of them allows a firing event for neurons whose membrane potentials surpass the threshold, i.e., $\Delta T = T_s \Delta t$. This equation bridges between a time-series time step $\Delta T$ and an SNN time step $\Delta t$. As a result, the independent variable $t$ in time-series ($\mathcal{X}(t)$) and in SNN ($U(t), I(t), H(t), S(t)$) are now sharing the same meaning.
> > > > > >
> > > > > > To this end, the spiking encoder, responsible for generating the first spike trains based on the floating-point inputs, needs to calculate $T_s \times T \times C$ possible spike events.
> > > > > > The most straightforward non-parametric approach is to consider each data point in the input time series as the current value and replicate it $T_s$ times.
> > > > > > However, this approach can disrupt the continuous nature of the underlying $\mathcal{X}(t)$ hypothesis.
> > > > > > Therefore, we seek to use parametric spike encoding techniques.
> > > > > >
> > > > > > **Convolutional Spike Encoder**: Given the historical observed time-series $\mathbf{X}\in\mathbb{R}^{T\times C}$, we feed it into a convolutional layer followed by batch normalization and generate the spikes as:
> > > > > > \begin{equation}
> > > > > > \mathbf{S} = \mathcal{SN}\left(\operatorname{BN}\left(\operatorname{Conv}\left(\mathbf{X}\right)\right)\right),
> > > > > > \end{equation}
> > > > > > where $\mathcal{SN}$ denotes a spiking neuron layer, by passing through the convolutional layer, the dimension of the spike train $\mathbf{S}$ is expanded to ${T_{s}\times T\times C}$. Spikes at every SNN time step are generated by pairing the data with different convolutional kernels. The convolutional spike encoder capture internal temporal information of the input data, i.e., temporal changes and shapes, respectively, contributing to the representation of the dynamic nature of the information over time and catering to the following spiking layers for event-driven modeling.
> > > > > >
> > > > > > A example for above description:
> > > > > >
> > > > > > Given a history inputs $X$ ={$x_1$, $x_2$, ..., $x_T$} $\in$ $R^{T \times D} $, we firstly use the **Convolutional Spike Encoder**  to transform $X$ into $\hat{X}$ $\in$ $R^{T_s \times T \times D}$, follow by the **batch normalization ($\operatorname{BN}$)** layer, and then input into a **spiking neuron layer**, which in our manuscript is **TS-LIF**.
> > > > > >
> > > > > > Q3: Why are additional $T_{s}$ time steps introduced instead of directly using  $T$ ?
> > > > > >
> > > > > > A3: We use $T_s$ instead of $T$, considering the following two points:
> > > > > >
> > > > > > (i) **Fair Comparison**:  To ensure a fair comparison between TS-LIF and the vanilla LIF structure in time series forecasting tasks, we followed the experimental setup in [3] and set the time steps to 4. This setting effectively highlights the advantages of TS-LIF in time series modeling.
> > > > > >
> > > > > > (ii) **Experimental Results**: We conducted experiments with two different $T_{s}$ settings to further demonstrate the effectiveness of incorporating additional $T_{s}$. Specifically, we set $T_{s}$ to **1**(without $T_{s}$ time steps) and **4** (with $T_{s}$ time steps) and performed comparative experiments with a prediction length of 24 on the Metr-la and Electricity datasets. The results are shown in the table below. We observed that when $T_{s}$ is set to 4, TS-LIF achieves performance improvements on both GRU and iTransformer. Therefore, our experimental setup includes the additional $T_{s}$ time steps.
> > > > > >
> > > > > > |Datasets|Metr-la| | | |Electricity| | | |
> > > > > > |-|-|-|-|-|-|-|-|-|
> > > > > > |$T_{s}$|1||4||1||4|
> > > > > > |Metrics|R²↑|RSE↓|R²↑|RSE↓|R²↑|RSE↓|R²↑|RSE↓|R²↑|RSE↓|R²↑|RSE↓|
> > > > > > |TS-GRU|0.609|0.663|**0.618**|**0.651**|0.975|0.253|**0.981**|**0.240**|
> > > > > > |TS-former|0.611|0.664|**0.620**|**0.655**|0.980|0.221|**0.985**|**0.215**|
> > > > > >
> > > > > > We sincerely hope our clarifications above have addressed your concerns and can improve your opinion of our work. If you have any further questions, we are looking forward to discussing with you.

---

> > > > > > > ### Comment · Reviewer_a5wF · 2024-11-26
> > > > > > >
> > > > > > > Thank you for the detailed explanations, which have addressed my concern regarding the experimental setup. As a result, I have increased my score to 6. I recommend including baseline results of existing spiking neuron models for all evaluated tasks in Table 1 to provide a more comprehensive comparison.

---

> > > > > > > > ### Author Response · Authors · 2024-11-26
> > > > > > > > **Response to Reviewer a5wF**
> > > > > > > >
> > > > > > > > Thank you for your positive feedback and raising the score. We are glad to hear that our rebuttal addressed your concerns regarding the experiment's setup. We will conduct comprehensive experiments on the spiking neuron models you mentioned in the coming period and incorporate them into the revised version of our TS-LIF.
> > > > > > > >
> > > > > > > > Thank you once again for your support and constructive feedback. We look forward to finalizing the manuscript.

---

### Official Review · Reviewer_nqfP · 2024-10-30

**Soundness:** 2
**Presentation:** 3
**Contribution:** 2
**Rating:** 6
**Confidence:** 4

**Summary:**

This article improves the dynamical characteristics of the LIF neuron and proposes a TS-LIF model for time series forecasting, incorporating interactions between dendrites and soma. The authors analyze the robustness and functionality of the improved dynamical system's different components and validate through experiments that TS-LIF outperforms the standard LIF on time series prediction benchmarks.

**Strengths:**

* The analysis of the improved model is quite thorough.
* Experiments demonstrate that it outperforms previous SNN-based methods.
* The writing is clear and easy to understand.

**Weaknesses:**

The main issue with this article is the somewhat unclear connection between the motivation and contributions. I believe the authors need to clarify their motivation for using a spiking model for time series tasks.

If the goal of using a spiking model is to achieve better performance, the following points should be addressed:

* If the spiking model is used to improve performance, it should be compared with the latest state-of-the-art methods, such as [1] and [2].
* In the introduction, it states, "In contrast, SNNs, with their event-driven and sparse computational architecture, can offer a more efficient solution, particularly for applications that involve sparse temporal events and demand low energy consumption." If the motivation for using a spiking model is to achieve low energy consumption, then additional experiments on hardware platforms should be provided to verify the model's energy efficiency, including specific tests on platforms like FPGA, GPU, and neuromorphic chips [3] to support this claim.
* The authors improved the model by introducing input stimuli into the soma, achieving a symmetric dynamical form. While this improvement indeed enhances performance, it also introduces additional computational overhead. I believe it would be beneficial to compare this model to others in terms of parameter count, training time, and FLOPs to illustrate the advantages of using a spiking model.

In the abstract, the authors mention that the spiking model is a biologically inspired neural network. If the purpose of this work is to validate the biological plausibility of this approach (or if the motivation is computational neuroscience rather than merely solving an engineering problem), then I believe the following points should be further addressed.

* In the proposed dynamic form where input stimuli are added to the soma rather than only to the dendrites, is there evidence that this mechanism is employed in the brain for similar time series forecasting processes?
* Can dynamics or data similar to TS-LIF be observed in the biological brain, thereby validating that TS-LIF indeed simulates brain mechanisms?
* It is suggested that the authors theoretically connect "shortcuts in soma" to analogous time series prediction processes in the brain, such as through predictive coding, active inference, etc. (This is optional, as it is challenging, but it would greatly enhance the contribution of the paper.)

[1] Lin, Shengsheng, et al. "SparseTSF: Modeling Long-term Time Series Forecasting with 1k Parameters." arXiv preprint arXiv:2405.00946 (2024).

[2] Ilbert, Romain, et al. "Unlocking the potential of transformers in time series forecasting with sharpness-aware minimization and channel-wise attention." arXiv preprint arXiv:2402.10198 (2024).

[3] Yao, Man, et al. "Spike-based dynamic computing with asynchronous sensing-computing neuromorphic chip." Nature Communications 15.1 (2024): 4464.

**Questions:**

Please see weaknesses above.

---

> ### Author Response · Authors · 2024-11-23
> **Response to Reviewer nqfP (Part 1)**
>
> Thank you for taking the time to review our paper and providing valuable feedback. We would like to answer your questions below.
>
> **W1: The motivation for using a spiking model for time series tasks**
>
> **WA1**: We aim to propose a novel spike neuron (TS-LIF) that aligns better with the characteristics of time series forecasting tasks. We would like to provide a more detailed explanation below:
> 1. Spiking Neural Networks (SNNs) are inherently designed to process temporally correlated data, making them naturally aligned with time series tasks due to their **event-driven** and **time-sensitive** properties.
> 2. SNNs excel at handling sparse data, as evidenced by their **robustness** in dealing with **missing values** in time series data, which is a common challenge in real-world scenarios (e.g., see Table 2 and Table 9).
> 3. Existing time series processing models, such as Transformer, often require significant computational resources. In contrast, the sparse firing characteristic of SNNs allows for a substantial reduction in the model's energy consumption. To support this, we have conducted additional experiments demonstrating the **energy efficiency** of our approach (e.g., refer to Table 6 of Appendix).
>
> As a result, TS-LIF is a novel spike neuron developed to explore the application of SNN frameworks in time series forecasting.
>
> **W2: Compared with the latest state-of-the-art time series methods (SparseTSF, SAMformer)**
>
> **WA2**: We conducted comparison experiments between TS-LIF and [1], [2] using three prediction lengths on two datasets (Metr-la and Electricity). SparseTSF* replaces the original ReLU activation in SparseTSF with TS-LIF, demonstrating TS-LIF's transferability and compatibility with ANN architectures, while SAMformer cannot be modified. Results show TS-former outperforms others on Metr-la, whereas SparseTSF* and SparseTSF perform slightly better than TS-former on Electricity. These findings confirm the compatibility of TS-LIF with ANN frameworks and its potential to enhance SNN frameworks in time series forecasting.
>
> |**Datasets**|**Metr-la**|||**Electricity**|||
> |-|-|-|-|-|-|-|
> |**Lengths**|**24**|**48**|**96**|**24**|**48**|**96**|
> |**Models**|**R²↑/RSE↓**|**R²↑/RSE↓**|**R²↑/RSE↓**|**R²↑/RSE↓**|**R²↑/RSE↓**|**R²↑/RSE↓**|
> |**SparseTSF**|0.576/0.681|0.427/0.792|0.253/0.916|**0.991/0.167**|**0.986/0.195**|**0.982/0.232**|
> |**SparseTSF***|0.580/0.692|0.426/0.801|0.247/0.924|0.990/0.177|0.985/0.201|**0.982/0.234**|
> |**SAMformer**|0.549/0.739|0.401/0.863|0.219/0.965|0.983/0.218|0.980/0.239|0.978/0.257|
> |**TS-former**|**0.620/0.655**|**0.445/0.763**|**0.283/0.874**|0.985/0.215|0.981/0.234|0.977/0.261|
>
> **Note**: Results are evaluated across 3 prediction lengths (24, 48, 96) on the Metr-la and Electricity datasets.
> - **SparseTSF***: Replaces the ReLU function of SparseTSF with TS-LIF.
> - **TS-former** represent the iTransformer with our TS-LIF.
>
> **W3: Energy consumption, parameter count, training time, and FLOPs**
>
> **WA3**: Based on [3], we measured energy consumption on 45nm hardware and compared the parameter sizes of our TS-LIF and LIF under three different backbone structures. The detailed results are presented below. Since TS-LIF introduces only a few additional parameters, just four learnable parameters: $\alpha, \beta, \gamma$, and $\kappa$ (where $\kappa$ is a vector related to the network's dimensions, while the others are scalars) of SNN structure, TS-LIF does not significantly increase the model size or training/inference speed compared with SNN structure.
>
> |**Model**|**Param(M)**|**OPs(G)**|**Energy(mJ)**|**Energy Reduction**|**Train/Infer Time (s)**|**R²**|
> |-|-|-|-|-|-|-|
> |**TCN**|0.460|0.14|0.64|-|21.34/11.47|**.973**|
> |**Spike-TCN**|0.461|0.15|0.23|63.60%↓|306.91/27.85|.963|
> |**TS-TCN**|0.465|0.19|0.25|60.93%↓|308.26/28.14|.971|
> |**GRU**|1.288|1.32|6.07|-|37.73/7.35|.972|
> |**Spike-GRU**|1.289|1.63|1.51|75.05%↓|235.46/10.05|.964|
> |**TS-GRU**|1.291|1.67|1.58|73.80%↓|246.23/9.78|**.981**|
> |**iTransformer**|1.634|2.05|9.47|-|7.24/6.38|.977|
> |**iSpikformer**|1.634|3.55|3.19|66.30%↓|49.84/8.69|.974|
> |**TS-former**|1.640|3.59|3.22|65.99%↓|50.36/8.72|**.985**|
>
> **Note** (Theoretic energy consumption of Electricity during the inference stage, and the results are sourced from [4], except the Train/Infer Time volume and the rows of TS-TCN, TS-GRU and TS-former):
> - "OPs" refers to SOPs in SNN and FLOPs in ANN.
> - "SOPs" denotes the synaptic operations of SNNs. (The definition can refer to **Appendix A.5**)
> - "FLOPs" denotes the floating point operations of ANNs.
> - "Train/Infer Time" denotes the training and inference time of 15784 training data and 5260 test data.

---

> > ### Author Response · Authors · 2024-11-23
> > **Response to Reviewer nqfP (Part 2)**
> >
> > **W4: Can similar dynamics to those observed in TS-LIF be found in the biological brain? Could the authors establish a theoretical connection between "shortcuts in soma" and known brain prediction processes?**
> >
> > **WA4:** We provide a detailed response for biological plausibility and theoretical connections of our TS-LIF model:
> > 1. In the **Hierarchical Temporal Memory** (HTM) neuron model, numerous synapses distributed along the dendrites function as pattern detectors that trigger NMDA spikes and depolarization at the soma [5]. This aligns closely with the TS-LIF model's approach, which leverages dendritic and somatic integration to process inputs across **diverse temporal patterns**. Furthermore, the TS-LIF model’s ability to generate both dendritic and somatic spikes is inspired by the biological capability of neurons to simultaneously process and predict signals, similar to the prediction mechanisms observed in HTM neurons.
> > 2. Regarding the **"shortcuts in soma"**, we interpret these as effective feature extractors that act as identity mappings for specific dendrites, enabling efficient information propagation without transformation. This mechanism mirrors certain dendritic branches in biological neurons that transmit unaltered signals directly to the soma. Shortcuts act as mechanisms for retaining core features across layers, contributing to the model's ability to make predictions while preserving critical information.
> > 3. Moreover, the dendrites and soma in our TS-LIF model can be aslo viewed as **two interacting units (or "neurons")**. These units process information at different frequencies while influencing one another, resembling mechanisms observed in sensory systems, such as auditory processing (inner hair cells tuned to different characteristic frequencies) and visual systems (S, M, and L-type cone cells tuned to different wavelengths).
> >
> > A more detailed biological explanation is provided in the revised manuscript under the **Temporal Segment LIF Neuron** section. The modifications have been highlighted to clarify this connection and further emphasize the conceptual similarities.
> >
> > We sincerely hope our clarifications above have addressed your concerns and can improve your opinion of our work.
> >
> > [1] Lin, Shengsheng, et al. "SparseTSF: Modeling Long-term Time Series Forecasting with 1k Parameters." arXiv preprint arXiv:2405.00946 (2024).
> >
> > [2] Ilbert, Romain, et al. "Unlocking the potential of transformers in time series forecasting with sharpness-aware minimization and channel-wise attention." arXiv preprint arXiv:2402.10198 (2024).
> >
> > [3] Yao M, Zhao G, Zhang H, et al. Attention spiking neural networks[J]. IEEE transactions on pattern analysis and machine intelligence, 2023, 45(8): 9393-9410.
> >
> > [4] Lv C, Wang Y, Han D, et al. Efficient and Effective Time-Series Forecasting with Spiking Neural Networks[C]//Forty-first International Conference on Machine Learning.
> >
> > [5] Hawkins J, Ahmad S. Why neurons have thousands of synapses, a theory of sequence memory in neocortex[J]. Frontiers in neural circuits, 2016, 10: 174222.

---

> > ### Author Response · Authors · 2024-11-26
> > **Kindly Request for Reviewer's Feedback**
> >
> > Dear reviewer nqfP,
> >
> > Thank you for your time and effort in providing such valuable feedback on our work. As the discussion period is nearing its end, we hope you’ve had the chance to go through our rebuttal. We believe it has helped clarify our work further, and if our responses have addressed your concerns, we would greatly appreciate it if you could kindly update your feedback to reflect this. We’re more than happy to continue the discussion if there are any remaining questions or points to address.
> >
> > Sincerely
> >
> > Authors

---

> > > ### Comment · Reviewer_nqfP · 2024-11-26
> > >
> > > Thank you for your detailed and thorough response to the concerns raised. Your clarifications and additional experiments have addressed many of the key points, and I will be raising my rating for this paper accordingly.

---

> > > > ### Author Response · Authors · 2024-11-26
> > > > **Response to Reviewer nqfP**
> > > >
> > > > Dear reviewer nqfP,
> > > >
> > > > Thank you for your thoughtful response and positive feedback. We truly appreciate the time and effort you’ve dedicated to reviewing our work and are glad that our clarifications and additional experiments have addressed your concerns. Your insights are valuable in improving our work.
> > > >
> > > > Sincerely,
> > > >
> > > > Authors

---

### Official Review · Reviewer_fwe4 · 2024-11-10

**Soundness:** 3
**Presentation:** 3
**Contribution:** 3
**Rating:** 6
**Confidence:** 3

**Summary:**

This paper introduces TS-LIF, a novel dual-compartment spiking neuron model for time series forecasting. The authors argue that traditional LIF models struggle to capture long-term dependencies and multi-scale temporal dynamics. Their proposed TS-LIF model incorporates dendritic and somatic compartments to process different frequency components, aiming to improve the accuracy and robustness of time series forecasting.

**Strengths:**

1. **Novelty**

The introduction of the TS-LIF model with its dual-compartment architecture (and signal decomposition), together its application to series forecasting is a novel contribution to the field of spiking neural networks .

2. **Placement**

The relationship of the work as respect to previous ones is well discussed.

3. **Performance Improvement**

The experimental results demonstrate clear improvements over traditional LIF-based SNNs and even some ANNs on standard benchmark datasets.

4. **Robustness**

The model shows good robustness in the face of missing data, which is a common challenge in real-world time series data.

**Weaknesses:**

1. **Limited Biological Plausibility? **

While the authors claim biological inspiration, the specific implementation of the dual-compartment model may not have direct biological correlates.
E.g. I like the idea of decomposing of the input signal in different frequencies. (Maybe Hierarchical temporal memory could referenced here). Can the authors discuss more on the biological plausibility of this phenomenon?

2. ** Over-Reliance on Benchmark Comparisons**

The paper heavily focuses on comparing TS-LIF with existing models on benchmark datasets. While this demonstrates performance improvements, it doesn't provide a deep understanding of the model's behavior. A more in-depth analysis of the model's inner workings, including how the dendritic and somatic compartments interact and contribute to the final prediction, would be beneficial.

It would be nice to some example of the network dynamics in response to data input, for different dataset, before and after training, while forecasting, populations analysis (such as the average power spectrum for different compartment over many realizations of the experiment).

**Questions:**

- Have you explored the computational efficiency of TS-LIF compared to traditional ANNs, particularly in terms of energy consumption?

- Are there specific types of time series data or forecasting tasks where TS-LIF might be particularly well-suited or, conversely, might struggle?
- How does the choice of the weighting parameter κ in the mixed spike output affect the model's performance? Is there an optimal way to determine this parameter?

- I suggest to include those references on spiking networks that are relevant for the topic. [1] is a biologically implementation of RFLO in spiking networks, [2] discuss learning temporal sequences leveraging multi-compartment neurons and local errors, [3] discuss learning in deep architectures thanks to multi-compartment neurons.

[1] Bellec, Guillaume, et al. "A solution to the learning dilemma for recurrent networks of spiking neurons." Nature communications 11.1 (2020): 3625.

[2] Capone, Cristiano, et al. "Beyond spiking networks: the computational advantages of dendritic amplification and input segregation." Proceedings of the National Academy of Sciences 120.49 (2023): e2220743120.

[3] Payeur, Alexandre, et al. "Burst-dependent synaptic plasticity can coordinate learning in hierarchical circuits." Nature neuroscience 24.7 (2021): 1010-1019.

---

> ### Author Response · Authors · 2024-11-23
> **Response to Reviewer fwe4**
>
> We thank you for recognizing the novelty and contributions of our paper and also for the positive feedback. We would like to answer your questions below.
>
> **W1: Could the authors elaborate on the biological plausibility, possibly referencing Hierarchical Temporal Memory?**
>
> **WA1:** We agree that the HTM neuron model can further support the biological plausibility of our TS-LIF neuron model [1].
> 1. **Dendritic spike**: In the HTM model, numerous synapses are distributed along dendrites, with small subsets of neighboring synapses acting as pattern detectors. Collectively, these thousands of synapses function as independent pattern detectors, where the detection of any pattern triggers an NMDA spike, leading to depolarization at the soma.
> 2. **Shortcut connection**: These patterns may represent sequences of bioelectrical signals across different frequencies, including identity mappings.
> 3. **Simultaneous spike**: HTM neurons are capable of making multiple simultaneous predictions, aligning with the TS-LIF model’s capability of simultaneously generating dendritic and somatic spikes.
> 4. **Other perspective**: The dendrites and soma in our TS-LIF model can be aslo viewed as two interacting units (or "neurons"). These units process information at different frequencies while influencing one another, resembling mechanisms observed in sensory systems, such as auditory processing (inner hair cells tuned to different characteristic frequencies) and visual systems (S, M, and L-type cone cells tuned to different wavelengths).
>
> A more detailed biological explanation is provided in the revised manuscript under the **Temporal Segment LIF Neuron** section. The modifications have been highlighted to clarify this connection and further emphasize the conceptual similarities.
>
> **W2: Average power spectrum of the network dynamics**
>
> **WA2**: The temporal frequency information of the original signal is notably influenced as it passes through the encoder in the network. To illustrate this, we have plotted the power spectrum of dendritic and somatic voltages in TS-LIF after the first encoder layer of TS-former. These spectrum highlight how dendritic and somatic compartments process distinct frequency components, showcasing their unique responses. The detailed frequency power spectrum are provided in the **Appendix (Fig. 5)**.
>
> **Q1: Energy consumption?**
>
> **A1**: We have carefully evaluated this aspect and present our findings in the table below. Our model demonstrates a significant reduction in energy consumption compared to traditional ANN counterparts.
>
> **Table.Theoretical Energy Consumption and Performance Comparison During the Inference Stage of the Electricity Dataset**
> |**Model**|**Param(M)**|**OPs(G)**|**Energy(mJ)**|**EnergyReduction**|**R²**|
> |-|-|-|-|-|-|
> |**TCN**|0.460|0.14|0.64|-|**.973**|
> |**TS-TCN(Ours)**|0.465|0.19|0.25|60.93%↓|.971|
> |**GRU**|1.288|1.32|6.07|-|.972|
> |**TS-GRU(Ours)**|1.291|1.67|1.58|73.80%↓|**.981**|
> |**iTransformer**|1.634|2.05|9.47|-|.977|
> |**TS-former(Ours)**|1.640|3.59|3.22|65.99%↓|**.985**|
> - "OPs" refers to SOPs in SNN and FLOPs in ANN.
> - "SOPs" denotes the synaptic operations of SNNs. (The definition can refer to **Appendix A.5**)
> - "FLOPs" denotes the floating point operations of ANNs.
>
>
> **Q2: Are there specific types of time series data or forecasting tasks where TS-LIF might be particularly well-suited or, conversely, might struggle?**
>
> **A2**: Based on the results in our main experiment table, where TS-former outperforms ANN models on the Solar and Electricity datasets, we hypothesize that TS-LIF excels in time series where high-frequency and low-frequency information can be clearly decomposed. In contrast, for more challenging datasets like Metr-la, TS-LIF may struggle to perform as effectively. This is a highly insightful point, and we will conduct experiments to validate our hypothesis if time permits.
>
>
> **Q3: The choice of the weighting parameter $\kappa$**
>
> **A3:** The weighting parameter $\kappa$ controls the trade-off between dendritic spikes and somatic spikes. Instead of predefining this parameter, we treat it as a learnable parameter, allowing the model to determine its optimal value through training on the dataset.
>
> **Q4: Include new references that are relevant for the topic.**
>
> **A4:** Thank you for your valuable suggestion to include additional relevant references. In the revised manuscript, we have incorporated new references to enrich the discussion and provide a more comprehensive context for our work.
>
> [1] Hawkins J, Ahmad S. Why neurons have thousands of synapses, a theory of sequence memory in neocortex[J]. Frontiers in neural circuits, 2016, 10: 174222.
>
> [2] Lv C, Wang Y, Han D, et al. Efficient and Effective Time-Series Forecasting with Spiking Neural Networks[C]//Forty-first International Conference on Machine Learning.
>
> We sincerely hope our clarifications above have addressed your concerns and can improve your opinion of our work.

---

> ### Author Response · Authors · 2024-11-26
> **Kindly Request for Reviewer's Feedback**
>
> Dear reviewer fwe4,
>
> Thank you for your time and effort in providing such valuable feedback on our work. As the discussion period is nearing its end, we hope you’ve had the chance to go through our rebuttal. We believe it has helped clarify our work further, and if our responses have addressed your concerns, we would greatly appreciate it if you could kindly update your feedback to reflect this. We’re more than happy to continue the discussion if there are any remaining questions or points to address.
>
> Sincerely
>
> Authors

---

> > ### Comment · Reviewer_fwe4 · 2024-11-26
> >
> > I thank the authors for answering my questions, I would like to confirm my score.

---

> > > ### Author Response · Authors · 2024-11-26
> > > **Response to Reviewer fwe4**
> > >
> > > Dear reviewer fwe4,
> > >
> > > We sincerely appreciate your thoughtful review and valuable feedback. Thank you for considering our responses and for taking the time to confirm your score. Should you have any additional questions or suggestions, please feel free to let us know.
> > >
> > > Sincerely,
> > >
> > > Authors

---

### Author Response · Authors · 2024-12-03
**Summary of Revisions**

We sincerely thank all the reviewers for their insightful feedback and valuable comments, which are instructive for us to improve our manuscript.

In this work, we introduce the Temporal Segment Spiking Neuron (TS-LIF) for multivariate time series forecasting, where dendrites and soma process high- and low-frequency information, respectively. This decoupled design aligns naturally with the intrinsic properties of time series. Notably, we observe that the temporal segment structure excels in modeling **long-term dependencies** and demonstrates **better robustness** in scenarios with missing values, while previous SNN structures generally fail.

We're pleased that the reviewers agree our paper **"Dual-compartment separately capture low- and high-frequency information is novel"** (Reviewer fwe4, a5wF), **"Experimental results demonstrate improvements, compared with previous SNN-based methods.** (Reviewer fwe4, nqfP), **"The paper is generally well-written and easy to understand."** (Reviewer nqfP, toyw).

The reviewers raised insightful and constructive concerns. The major revisions we made during the rebuttal period are summarized as follows:

- **Biological Plausibility Discussion** (All Reviewers): We provided more discussion on biological plausibility of our TS-LIF, establishing connections with the **Hierarchical Temporal Memory (HTM) model** and **visual and auditory system**.
- **Technical Contributions** (Reviewer nqfP, toyw): TS-LIF excels in modeling long-term dependencies, offers greater robustness in handling missing values, and is more energy-efficient than ANN models.
- **Comprehensive Evaluations** (Reviewer a5wF): We have incorporated the requested evaluations, including comparisons with various SNN neurons and efficient time series models.

We sincerely appreciate the valuable suggestions from all reviewers, which have been greatly helpful in improving our manuscript.

All the best,

Authors

---

### Meta-Review · Area_Chair_xBVo · 2024-12-21

**Metareview:**

This paper describes TS-LIF mode, an extension to the leaky integrate-and-fire (LIF), which incorporates interactions between dndrites and soma in order to capture long-term dependencies in time series forecasting. The reviewers raised concerns about biological realism, efficiency, and rigorousness of the model comparisons, but were persuaded by the authors' thorough and detailed set of responses and additional performance comparisons. Ultimately, the reviewers were unified in appraising the work as above threshold for acceptance, concluding that the work was interesting, well-written, and represents a substantial advance on an important problem of interest. I'm pleased to report that it has been accepted to ICLR.  Congratulations!  Please revise the manuscript to address all reviewer comments and discussion points where possible.

**Additional Comments On Reviewer Discussion:**

The authors provided very lengthy and detailed replies to reviewer concerns about biological realism and comparisons to other methods. Ultimately, the reviewers were persuaded by these additional comparisons.  Although uniform 6s still makes the paper only a "weak accept", I felt that the fact that all four reviewers felt the paper was above threshold, and the fact that the authors did such a careful and thorough job showing comparisons to previous methods, that the paper warrants acceptance.

---

### Decision · Program_Chairs · 2025-01-22

Accept (Poster)